# Endothelial Unc5B controls blood-brain barrier integrity

Kevin Boyé [1], Luiz Henrique Geraldo [1,2], Jessica Furtado[1], Laurence Pibouin-Fragner[2], Mathilde Poulet [1,2], Doyeun Kim[3], Bryce Nelson[4], Yunling Xu[2], Laurent Jacob[2], Nawal Maissa[2], Dritan Agalliu [5], Lena Claesson-Welsh [6], Susan L. Ackerman[7] & Anne Eichmann[1,2,8✉]

Blood-brain barrier (BBB) integrity is critical for proper function of the central nervous system (CNS). Here, we show that the endothelial Unc5B receptor controls BBB integrity by maintaining Wnt/β-catenin signaling. Inducible endothelial-specific deletion of Unc5B in adult mice leads to BBB leak from brain capillaries that convert to a barrier-incompetent state with reduced Claudin-5 and increased PLVAP expression. Loss of Unc5B decreases BBB Wnt/β-catenin signaling, and β-catenin overexpression rescues *Unc5B* mutant BBB defects. Mechanistically, the Unc5B ligand Netrin-1 enhances Unc5B interaction with the Wnt co-receptor LRP6, induces its phosphorylation and activates Wnt/β-catenin downstream signaling. Intravenous delivery of antibodies blocking Netrin-1 binding to Unc5B causes a transient BBB breakdown and disruption of Wnt signaling, followed by neurovascular barrier resealing. These data identify Netrin-1-Unc5B signaling as a ligand-receptor pathway that regulates BBB integrity, with implications for CNS diseases.

[1] Cardiovascular Research Center, Department of Internal Medicine, Yale University School of Medicine, New Haven, CT, USA. [2] Université de Paris, INSERM, PARCC, F-75015 Paris, France. [3] OliX pharmaceuticals, Suwon, Republic of Korea. [4] Department of Pharmacology, Cancer Biology Institute, Yale University School of Medicine, New Haven, CT, USA. [5] Departments of Neurology and Pathology and Cell Biology, Columbia University Irving Medical Center, New York, NY, USA. [6] Department of Immunology, Genetics and Pathology, Uppsala University, Uppsala, Sweden. [7] Division of Biological Sciences Section of Neurobiology and Department of Cellular and Molecular Medicine, University of California San Diego and Howard Hughes Medical Institute, La Jolla, CA, USA. [8] Department of Molecular and Cellular Physiology, Yale University School of Medicine, New Haven, CT, USA. ✉email: anne.eichmann@yale.edu

The BBB protects the brain from toxins and pathogens and maintains homeostasis and proper function of the CNS[1,2]. To form the BBB, endothelial cells (ECs) lining brain blood vessels acquire a series of features that control the movement of ions, molecules and cells between the blood and the brain[3]. CNS ECs express specialized tight junctions that prevent paracellular exchange of small molecules[4,5]. They lack fenestrae and exhibit a low rate of transcytosis that limits transcellular exchange of large molecules[6,7], and they express transporters that shuttle essential nutrients into and out of the brain[8–10]. ECs acquire BBB properties through their interactions with mural cells and glial cells in the CNS microenvironment and are modulated by neuronal activity[11–14].

The canonical Wnt/β-catenin signaling pathway is a key regulator of BBB development and maintenance. CNS specific Wnt7a,7b and Norrin ligands produced by glial cells bind to multiprotein receptor complexes including Frizzled4 and LRP6 on brain ECs[15–21]. Receptor activation causes β-catenin stabilization and nuclear translocation to induce activation of TCF and LEF1 transcription factors, which control expression of a BBB-specific gene expression repertoire by inducing tight junction, solute transporter and efflux transporter expression, and by repressing expression of the permeability protein PLVAP that forms the diaphragm in EC fenestrae, caveolae and transcytotic vesicles[15,22–24]. Whether Wnt/β-catenin signaling could be modulated to open the BBB "on-demand" or to restore its integrity when damaged, is unknown.

Unc5 was initially discovered in a *C. elegans* screen for motor dysfunctions, and controls axon guidance in all species examined[25,26]. Among the four vertebrate Unc5 family members, only Unc5B is expressed in ECs in mice and humans[27]. Unc5B binds Netrin-1[26,28], Robo4[29,30] and Flrt2[31,32] via its extracellular domain (ECD). Unc5B signaling is mediated by its intracellular domain (ICD), which encompasses a membrane-proximal ZU5 domain (named for its homology to ZO1), followed by a UPA domain (named for its conservation in Unc5B, PIDD and Ankyrin[33]) and a death domain (DD) that mediates apoptosis in the absence of ligand[33,34]. These domains form a supramodule in which ZU5 binds to both UPA and DD suppressing Unc5B biological function, while ligand binding to Unc5B triggers a conformational change such that ZU5 loses its interaction with DD and exposes the UPA domain to activate Unc5B signaling[33]. Global homozygous *Unc5B* knockout in mice is embryonically lethal due to placental vascular defects[27,31], demonstrating that Unc5B has important functions in vascular development. To address biological roles of this receptor during postnatal and adult life, we generated mice with temporally inducible, endothelial-specific Unc5B deletions.

Here, we identify the endothelial transmembrane Unc5B receptor and its ligand Netrin-1 as regulators of the BBB integrity. Inducible endothelial-specific Unc5B deletions in adult mice leads to reduced Wnt/β-catenin signaling and BBB leak for tracers up to 40 kDa. *Unc5B* and *β-catenin* genetically interact in ECs to maintain BBB integrity and overexpressing an activated form of β-catenin rescues BBB defects induced by loss of Unc5B. We show that intravenous delivery of monoclonal antibodies blocking Netrin-1 binding to Unc5B induce transient BBB opening to bioactive molecules, which could be useful for drug delivery in various neurological diseases.

## Results

**Endothelial Unc5B controls BBB development and maintenance.** We generated *Unc5B^{fl/fl}* mice (Supplementary Fig. 1a) that were intercrossed with *Cdh5Cre^{ERT2}* mice (hereafter *Unc5BiECko*), which produces EC-specific deletion[35]. Gene deletion was induced by Tamoxifen (TAM) injection in neonates between postnatal day (P)0-P2, and qPCR revealed efficient *Unc5B* deletion (Supplementary Fig. 1b, c). Interestingly, neonatal TAM injection-induced seizures and lethality of *Unc5BiECko* mice around P12 (Supplementary Fig. 1d, Supplementary movie), indicating a possible BBB failure[2]. Intraperitoneal injection of a fluorescent tracer cadaverine (MW 950 Da) into P5 mice and analysis of tracer leak 2 h later revealed widespread tracer extravasation into the brain of P5 *Unc5BiECko* mice, confirming that *Unc5B* deletion impaired BBB development (Supplementary Fig. 1e).

To determine if Unc5B also controlled BBB integrity in adults, we induced gene deletion in two months old mice and probed BBB integrity 7 days later by intravenous (i.v.) cadaverine injection (Fig. 1a). Mice were sacrificed 30 min after dye injection and brain sections were immunolabeled with an antibody against podocalyxin, which labels the luminal EC membrane[36]. Cadaverine remained inside the vasculature of TAM-injected Cre-negative littermate controls but leaked into the *Unc5BiECko* brain (Fig. 1b). Quantification of cadaverine extravasation (see methods) revealed significantly increased leak across the BBB in *Unc5BiECko* brains when compared to controls, while the vascular permeability to cadaverine in other *Unc5BiECko* organs was similar to controls (Fig. 1c), indicating that Unc5B has a CNS-selective BBB-protective function in adult mice.

Staining of adult brain sections with a commercial antibody recognizing Unc5B showed labeling of endothelium in various brain regions and revealed that adult Unc5B deletion had no effect on vascular density over the period observed (Fig. 1d, e) Widespread but non-uniform cadaverine leakage was observed in several brain regions of *Unc5BiECko* mice, including the retrosplenial and piriform cortex, hippocampus, hypothalamus, thalamus, striatum and cerebellum, while other cortical areas such as the posterior parietal association areas and the primary somatosensory cortex displayed less cadaverine leak (Fig. 1f). Unc5B expression in cortical ECs of wildtype mice was similar between areas that displayed more or less cadaverine leak in *Unc5BiECko* brains (Fig. 1g), hence Unc5B expression alone was not sufficient to predict severity of BBB breakdown.

Unc5B expression was also detected in some CD13[+] pericytes (Fig. 1h). Because pericytes contribute to BBB integrity[11,12], we determined whether pericyte-derived Unc5B affected the BBB by crossing the *Unc5B^{fl/fl}* mice with *PdgfrβCre^{ERT2}* mice[37] (*Unc5BiPCko*), to delete *Unc5B* in mural cells. Neither TAM-treated Cre-negative littermate controls, nor *Unc5BiPCko* mice showed cadaverine leakage across the BBB (Fig. 1i, j). Hence, endothelial, but not pericyte, Unc5B controls adult BBB integrity.

**Endothelial Unc5B controls Claudin-5 and PLVAP expression.** To determine the cause of the BBB defect, we examined the expression levels of proteins relevant for BBB function in ECs, pericytes and astrocytes by western blot and immunolabeling[5,11–13,38]. Compared to TAM-treated Cre-negative littermate controls, *Unc5BiECko* brains had similar expression of EC BBB regulators such as Caveolin-1, ZO1, and Occludin (Fig. 2a–c). In addition, vessel coverage by PDGFRβ+ pericytes, and GFAP and Aquaporin-4 expression in astrocytes and astrocyte endfeet were also similar between genotypes (Fig. 2a–e). By contrast, *Unc5BiECko* mice showed significantly reduced expression of Claudin-5, along with increased expression of PLVAP in Western blot analysis of brain lysates (Fig. 2f, g). Immunolabeling of adult brain sections showed decreased Claudin-5 and increased PLVAP expression in areas with cadaverine leakage in *Unc5BiECko* brains (Fig. 2h), suggesting that BBB leakage may be due to changes in expression of Claudin-5

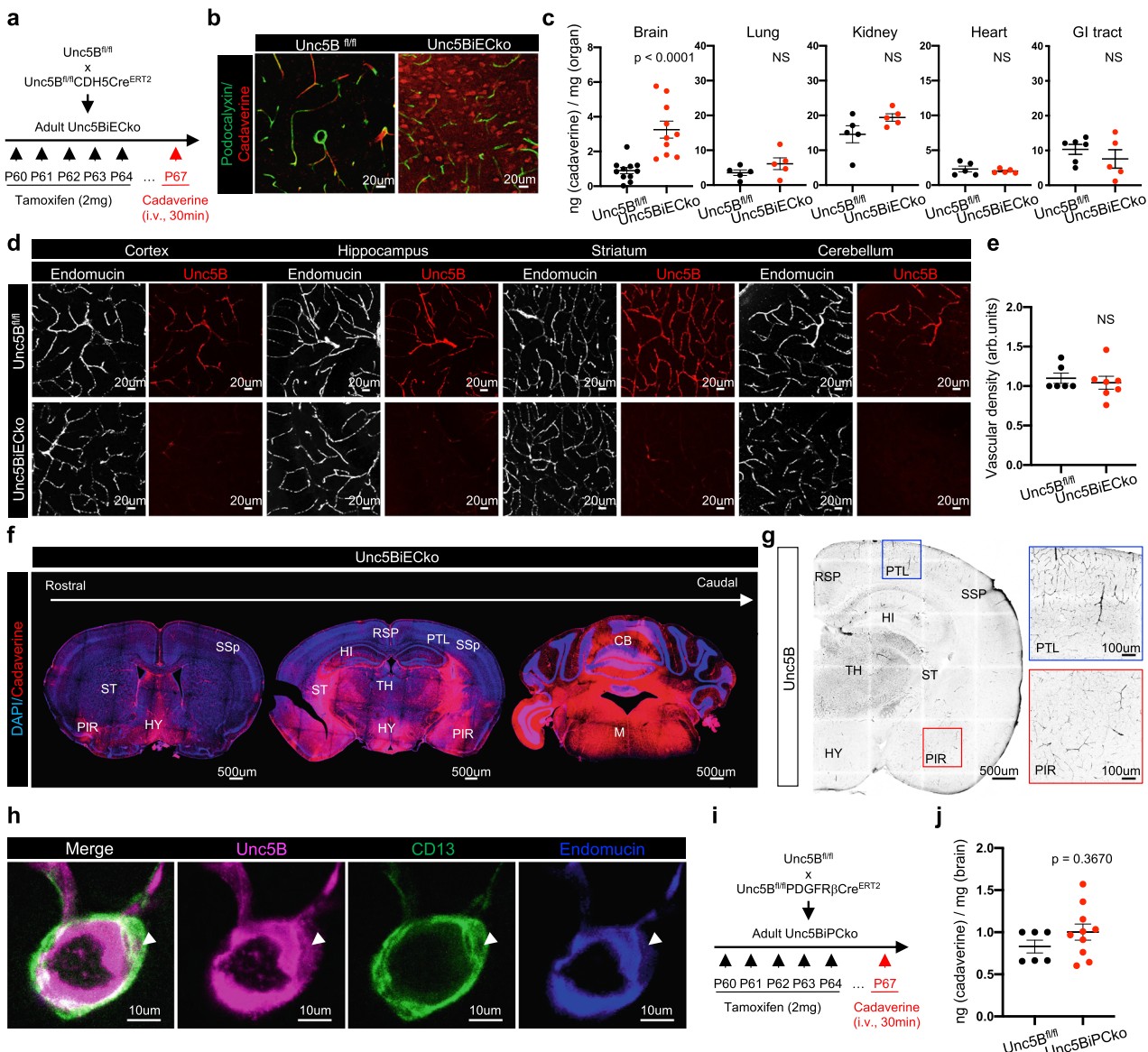

**Fig. 1 Endothelial Unc5B controls BBB integrity. a** *Unc5B* gene deletion strategy using tamoxifen injection in adult mice. **b** Immunofluorescence staining with the indicated antibodies and confocal imaging of adult brain sections at P67, 30 min after i.v cadaverine injection and reproduced on *n* = 4 *Unc5B^{fl/fl}* and *n* = 5 *Unc5BiECko* brains. **c** Quantification of dye content in brains and peripheral organs at P67, 30 min after i.v. cadaverine injection, *n* = 11 *Unc5B^{fl/fl}* and *n* = 10 *Unc5BiECko* brains (exact *p*-value = 0.0000397); *n* = 5 *Unc5B^{fl/fl}* and *n* = 10 *Unc5BiECko* lungs, kidneys and hearts; *n* = 6 *Unc5B^{fl/fl}* and *n* = 5 *Unc5BiECko* GI tracts. Each dot represents one mouse. **d** Immunofluorescence staining with the indicated antibodies and confocal imaging of adult brain sections, reproduced on *n* = 4 *Unc5B^{fl/fl}* and *n* = 4 *Unc5BiECko* brains. **e** Quantification of endomucin+ vascular density. Each dot represents the mean of several images, *n* = 6 *Unc5B^{fl/fl}* and *n* = 7 *Unc5BiECko* brains. One control mouse value was set as 1. **f** Tile-scan confocal imaging of adult *Unc5BiECko* brain sections at P67, 30 min after i.v cadaverine injection and reproduced on *n* = 3 *Unc5B^{fl/fl}* and *n* = 3 *Unc5BiECko* brains. **g** Immunofluorescence staining of Unc5B and tile-scan confocal imaging of brain sections, reproduced on *n* = 4 mice. Boxes show higher magnifications of cortical areas. **h** Immunofluorescence staining with the indicated antibodies and confocal imaging of adult brain sections, reproduced on *n* = 4 *Unc5B^{fl/fl}* and *n* = 4 *Unc5BiECko* brains. Arrowhead: Unc5B+/CD13+ pericyte. **i, j** *Unc5B^{fl/fl}* was crossed with *PDGFRβCre^{ERT2}* and BBB permeability was assessed at P67, 30 min after i.v. cadaverine injection, *n* = 6 *Unc5B^{fl/fl}* and *n* = 10 *Unc5BiPCko* brains. Each dot represents one mouse. All data are shown as mean ± SEM. NS non-significant, RSP Retrosplenial cortex, PTL Posterior parietal association areas, SSp Primary somatosensory cortex, PIR Piriform cortex, HI Hippocampus, HY Hypothalamus, TH Thalamus, ST Striatum, CB Cerebellum, M Medulla. Two-sided Mann–Whitney *U* test was performed for statistical analysis. Source data are provided as a Source Data file.

and PLVAP. Conversion of ECs to a Claudin-5-negative, PLVAP-positive state occurred in *Unc5BiECko* capillaries but not in larger vessels (>10 um in diameter, Fig. 2h, Supplementary Fig. 2a), suggesting that BBB leakage in *Unc5BiECko* brains originates from capillaries. To validate *Unc5B* deletion effects on Claudin-5 and PLVAP in an independent knockout mouse strain, we tested global *Unc5B* KO embryos at E12.5[27], which confirmed decreased *CLDN5* mRNA and protein levels as well as increased PLVAP

protein expression in homozygous mutant brain ECs when compared to wildtype control littermate brains (Supplementary Fig. 2b–e).

**Unc5B regulates Wnt/β-catenin signaling.** Because Claudin-5 and PLVAP are two known targets of CNS Wnt/β-catenin signaling[22–24], we determined if Unc5B affected Wnt signaling at the BBB. qPCR on adult brain lysates revealed decreased mRNA

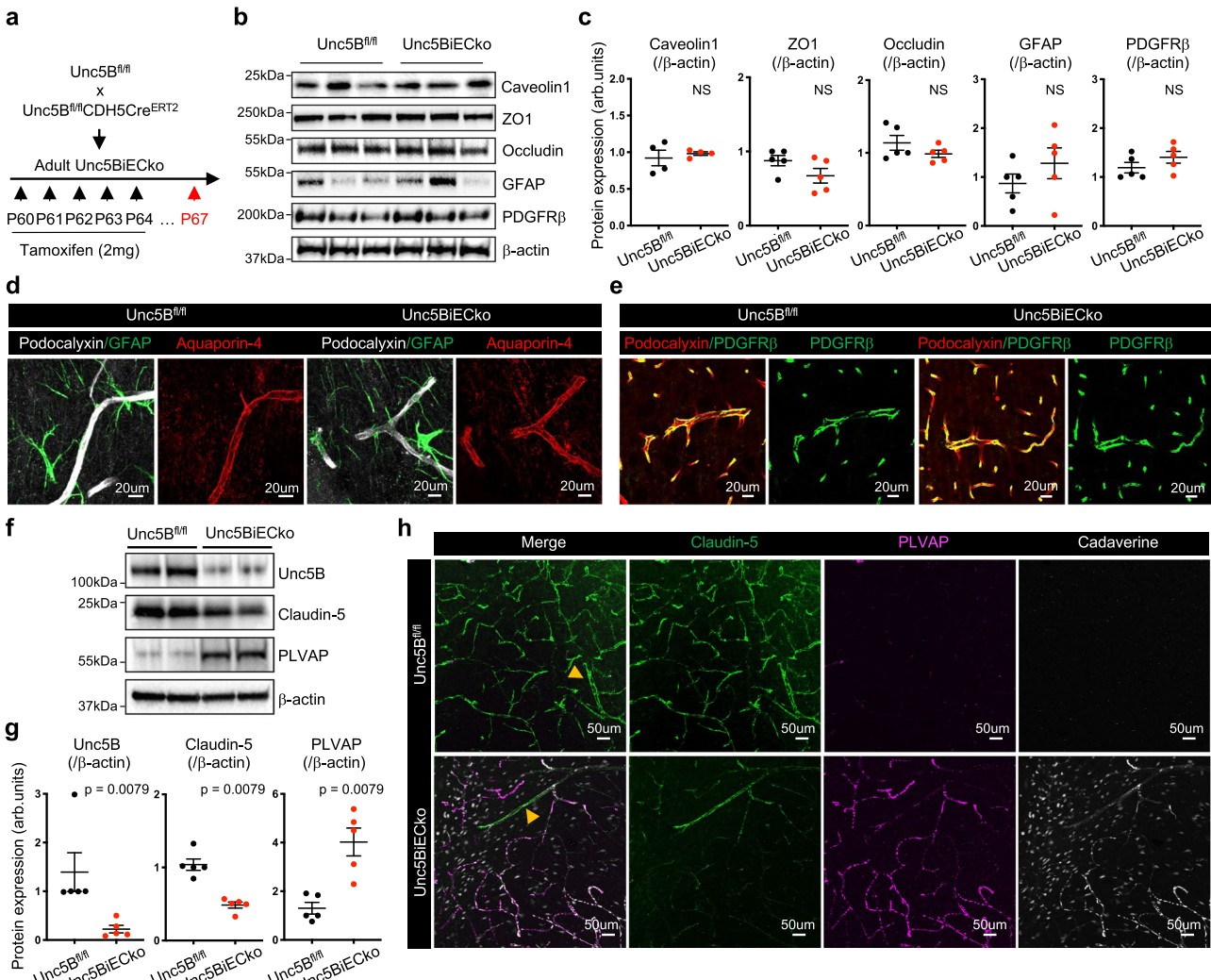

**Fig. 2 Unc5B controls Claudin-5 and PLVAP expression. a** *Unc5B* gene deletion strategy using tamoxifen injection in adult mice. Western blot (**b**) and quantification (**c**) of brain protein extracts at P67. Quantification of Caveolin-1 expression was performed on $n = 4$ *Unc5B*$^{fl/fl}$ and $n = 4$ *Unc5BiECko* brains. Quantification of ZO1, Occludin, GFAP and PDGFRβ expression was performed on $n = 5$ *Unc5B*$^{fl/fl}$ and $n = 5$ *Unc5BiECko* brains. Each dot represents one mouse. One control mouse was set as 1. **d**, **e** Immunofluorescence staining with the indicated markers and confocal imaging of brain sections, reproduced on $n = 4$ *Unc5B*$^{fl/fl}$ and $n = 4$ *Unc5BiECko* brains. Western blot (**f**) and quantification (**g**) of P67 brain protein extracts, $n = 5$ *Unc5B*$^{fl/fl}$ and $n = 5$ *Unc5BiECko* brains. Each dot represents one mouse. One control mouse value was set as 1. **h** Immunofluorescence staining with the indicated antibodies and confocal imaging of P67 piriform cortex 30 min after i.v cadaverine injection, reproduced on $n = 4$ *Unc5B*$^{fl/fl}$ and $n = 4$ *Unc5BiECko* brains. Arrowheads: larger vessels. All data are shown as mean ± SEM. NS non-significant. Two-sided Mann–Whitney *U* test was performed for statistical analysis. Source data are provided as a Source Data file.

levels of *LEF1* and *CLDN5* as well as increased *PLVAP* mRNA levels in *Unc5BiECko* brains compared to Cre-negative littermate controls (Fig. 3a). Next, we crossed *Unc5BiECko* mice with *TCF/LEF:H2B-GFP* mice[39] which express a GFP reporter of β-catenin transcriptional activity. Compared to TAM-treated Cre-negative controls, *Unc5BiECko;TCF/LEF:H2B-GFP* brains revealed decreased GFP expression in ECs nuclei labeled with the endothelial transcription factor ERG[40], attesting to decreased endothelial β-catenin transcriptional activity upon loss of Unc5B function (Supplementary Fig. 3a–c). Western-blot analysis on brain lysates revealed a significant decrease of β-catenin and LEF1 protein in *Unc5BiECko* brains compared to Cre-negative littermate controls (Fig. 3b, c). Moreover, phosphorylation of LRP6 at S1490, a hallmark of Wnt/β-catenin pathway activation[41], was dramatically downregulated upon *Unc5B* gene deletion (Fig. 3b, c). This phosphorylation provides a docking site for the adapter protein Axin1, resulting in inhibition of the β-catenin destruction complex and thereby promoting β-catenin nuclear translocation and activation[41,42]. Immunostaining confirmed decreased LEF1 expression in brain ECs of *Unc5BiECko* mice (Fig. 3d, e).

Immunoprecipitation of Unc5B from primary microvascular mouse brain ECs pulled down LRP6, pLRP6, Frizzled4 and the Wnt co-receptor GPR124 (Fig. 3f), demonstrating a physical interaction between Unc5B and Wnt receptors. To determine which Unc5B domain mediated this interaction, we infected Unc5B siRNA-treated human ECs with GFP-tagged siRNA resistant rat adenoviral constructs encoding Unc5B full-length (FL) or a cytoplasmic domain deletion (ΔCD) (Fig. 3g). LRP6 co-IP was rescued by Unc5B FL but not by ΔCD, identifying the Unc5B cytoplasmic domain as the main LRP6 interacting domain. Additional cytoplasmic domain deletions revealed that the Unc5B death domain was dispensable for LRP6 interaction, whereas deletion of the UPA domain abolished LRP6 co-IP. A construct encoding only the cytoplasmic UPA domain was

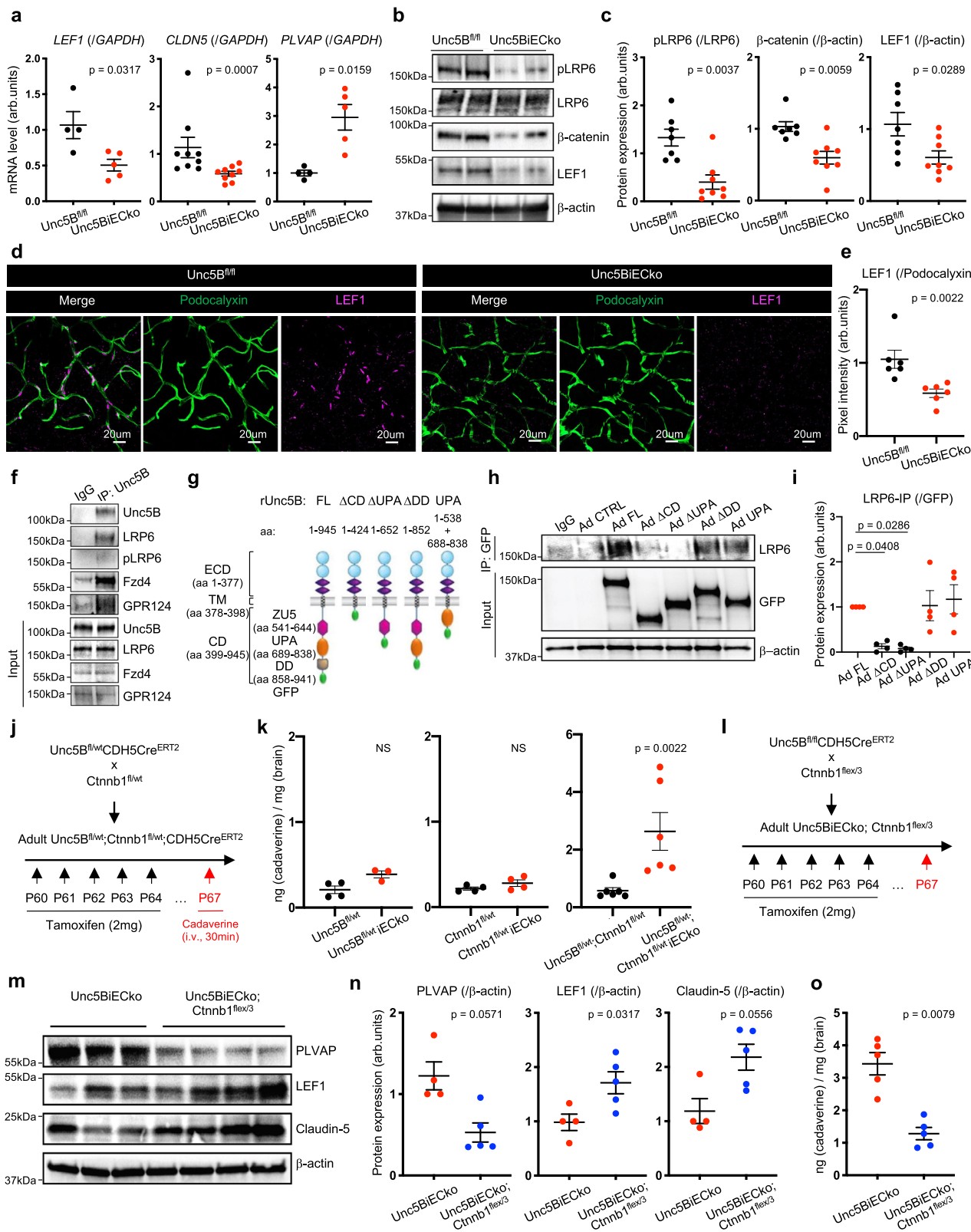

sufficient to rescue LRP6 co-IP (Fig. 3h, i), indicating that Unc5B interacts with LRP6 via its UPA domain.

To test genetic interaction of Unc5B with Wnt/β-catenin signaling, we generated a TAM-inducible endothelial-specific β-catenin deletion by crossing *Ctnnb1fl/fl* mice[43] with *CDH5CreERT2* mice (Fig. 3j, Supp. Figure 3d–f). Heterozygous *Unc5Bfl/wt-CDH5CreERT2* or *Ctnnb1fl/wt-CDH5CreERT2* mice displayed no

BBB cadaverine leakage, but BBB cadaverine leakage increased in double heterozygous *Unc5Bfl/wt;Ctnnb1fl/wt-CDH5CreERT2* brains compared to TAM-treated Cre-negative controls (Fig. 3j, k), demonstrating that *Unc5B* and *β−catenin* genetically interact in ECs to maintain BBB integrity.

Next, we crossed *Unc5BiECko* with mice overexpressing a TAM-inducible activated form of β-catenin (*Ctnnb1flex/3* mice[44]),

**Fig. 3 Unc5B regulates BBB Wnt/β-catenin signaling. a** qPCR analysis of P67 brain mRNA extracts, $n = 4$ $Unc5B^{fl/fl}$ and $n = 5$ $Unc5BiECko$ brains for quantification of *LEF1* and *PLVAP* mRNA levels, $n = 9$ $Unc5B^{fl/fl}$ and $n = 9$ $Unc5BiECko$ brains for quantification of *CLDN5* mRNA levels. Each dot represents one mouse. One control mouse was set as 1. Western blot (**b**) and quantification (**c**) of Wnt/β-catenin signaling components in brain protein extracts, $n = 7$ $Unc5B^{fl/fl}$ and $n = 8$ $Unc5BiECko$ brains. Each dot represents one mouse. One control mouse was set as 1. Immunofluorescence (**d**) and quantification (**e**) of LEF1 staining on adult brain sections. Each dot is the mean of several images, $n = 6$ $Unc5B^{fl/fl}$ and $n = 6$ $Unc5BiECko$ brains. One control mouse was set as 1. **f** CTRL IgG and Unc5B immunoprecipitation on cultured brain endothelial cells, reproduced on $n = 3$ independent experiment. **g** Schematic of Unc5B adenoviral constructs. CTRL IgG or GFP immunoprecipitation in Unc5B siRNA knockdown ECs infected with siRNA resistant Unc5B adenovirus (**h**), and quantification of LRP6 pulldown (**i**), $n = 4$ independent experiment. Each dot represents one independent experiment. **j** *Unc5B* and *Ctnnb1* gene deletion strategy using tamoxifen injection in adult mice. **k** Quantification of cadaverine content in P67 brains, 30 min after i.v. cadaverine injection, $n = 4$ $Unc5B^{fl/wt}$, $n = 3$ $Unc5B^{fl/wt}iECko$, $n = 4$ $Ctnnb1^{fl/wt}$, $n = 4$ $Ctnnb1^{fl/wt}iECko$, $n = 6$ $Unc5B^{fl/wt};Ctnnb1^{fl/wt}$ and $n = 6$ $Unc5B^{fl/wt};Ctnnb1^{fl/wt}iECko$ brains. Each dot represents one mouse. **l** *Unc5B* gene deletion and *Ctnnb1^{flex/3}* gene overexpression strategy using tamoxifen injection. Western blot (**m**) and quantification (**n**) of P67 brain protein extracts, $n = 4$ $Unc5BiECko$ and $n = 5$ $Unc5BiECko;Ctnnb1^{flex/3}$ brains. Each dot represents one mouse. One control mouse was set as 1. **o** Quantification of P67 brain cadaverine content, 30 min after i.v cadaverine injection, $n = 5$ $Unc5BiECko$ and $n = 5$ $Unc5BiECko;Ctnnb1^{flex/3}$ brains. Each dot represents one mouse. All data are shown as mean ± SEM. NS non-significant. Two-sided Mann–Whitney *U* test was performed for statistical analysis between two groups. ANOVA followed by Bonferroni's multiple comparisons test was performed for statistical analysis between multiple groups. Source data are provided as a Source Data file.

thereby enhancing endothelial Wnt/β-catenin signaling[22,24,45] (Fig. 3l, Supplementary Fig. 3g–i). The resulting offspring (*Unc5BiECko; Ctnnb1^{flex/3}*) displayed decreased PLVAP protein expression, along with increased LEF1 and Claudin-5 protein expression compared to *Unc5BiECko* mice (Fig. 3m, n). Cadaverine injection into *Unc5BiECko;Ctnnb1^{flex/3}* mice showed that BBB leakage was reduced by β-catenin overexpression in *Unc5BiECko* mice (Fig. 3o). Western-blot on brain lysates from adult TAM-injected *Ctnnb1^{fl/fl};CDH5Cre^{ERT2}* and *Ctnnb1^{flex/3}; CDH5Cre^{ERT2}* mice showed that loss or gain of endothelial β-catenin did not significantly change Unc5B or Netrin-1 expression (Supplementary Fig. 3d–i), suggesting that Unc5B acted upstream of Wnt signaling.

**Claudin-5 but not Vegfr2 is required for Unc5B BBB regulation.** To test if reduced Claudin-5 expression in *Unc5BiECko* brains was functionally involved in the *Unc5BiECko* BBB defect, we crossed *Unc5BiECko* mice with *eGFP::Claudin-5* mice that express 2-fold higher Claudin-5 levels compared to wildtype littermates and thereby display enhanced CNS EC paracellular barrier properties[46]. BBB leakage of cadaverine into the brains of *Unc5BiECko; eGFP::Claudin-5* mice was reduced compared to *Unc5BiECko* mice (Fig. 4a, b), demonstrating that increasing Claudin-5 levels rescued leakage of small MW tracers in Unc5B mutant brain ECs.

We considered other signaling pathways that could contribute to BBB leakage in *Unc5BiECko* mice. Unc5B inhibits Vegfr2-mediated permeability signaling in ECs in vitro by reducing phosphorylation of the Y949 residue[30]. Y949 phosphorylation is known to trigger disassembly of adherens junctions by activating VE-cadherin phosphorylation, which then downregulates Claudin-5[29,30,47]. Therefore, increased brain Vegfr2-Y949 permeability signaling in the absence of Unc5B could contribute to BBB opening. Western blotting of brain lysates revealed increased Vegfr2-Y949 phosphorylation in *Unc5BiECko* compared to Cre-negative littermate controls, while Vegfr2-Y1173 phosphorylation, which is critical for VEGF-induced proliferation, was unaffected (Fig. 4c, d). To test Vegfr2-Y949 function, we crossed *Unc5BiECko* mice with *Vegfr2-Y949F* mutant mice, which carry an inactivating substitution of tyrosine to phenylalanine and are resistant to VEGF-induced permeability[48]. Injection of fluorescent cadaverine revealed increased dye leakage into the brain of *Unc5BiECko; Y949F* mice compared to Cre-negative littermate controls (Fig. 4e, f), demonstrating that Vegfr2-Y949F failed to rescue BBB integrity in Unc5B mutant mice. Moreover, similar VE-cadherin expression and junctional coverage were observed in CTRL and *Unc5BiECko* brains, further attesting that Unc5B

regulation of BBB integrity is independent of the Vegfr2 VE-cadherin pathways (Fig. 4g–j).

**Netrin-1 controls BBB integrity.** We determined whether Unc5B ligands Netrin-1 and Robo4 regulated BBB integrity and Wnt/β-catenin pathway activation in CNS ECs. Since Netrin-1 mRNA is produced by several cell types in the adult brain[49], we generated temporally inducible *Netrin-1* global KO mice by crossing *Ntn1^{fl/fl}* mice[50] with *RosaCre^{ERT2}* mice (hereafter *Ntn1iko*), to induce ubiquitous gene deletion upon TAM injection. Compared to TAM-treated Cre-negative littermate controls, i.v. injection of cadaverine in adult *Ntn1iko* mice revealed increased cadaverine leakage across the BBB (Fig. 5a), while *Robo4* KO mice[29] did not exhibit any BBB leakage (Fig. 5b). Further analysis of adult *Ntn1iko* mouse brain lysates revealed efficient *Ntn1* gene deletion along with decreased pLRP6, Claudin-5 and LEF1 protein expression, while PLVAP expression was increased (Fig. 5c, d). Moreover, treating serum-starved mouse primary brain ECs with Netrin-1 increased LRP6 phosphorylation with a peak at 30 min to 8 h after stimulation (Fig. 5e, f). This effect was abolished by *Unc5B* siRNA treatment (Fig. 5e, f). Unc5B immunoprecipitation from mouse brain lysates revealed reduced LRP6 co-IP in the *Ntn1iko* mice when compared to controls (Fig. 5g, h), suggesting that Netrin-1 binding to Unc5B regulated LRP6 phosphorylation and Wnt/β-catenin activation in CNS ECs. We reasoned that Netrin-1 could modulate LRP6 phosphorylation via FAK, a kinase that regulates β-catenin in pluripotent embryonic stem cells[51]. Netrin-1-treated mouse brain ECs showed increased FAK phosphorylation from 1 to 8 h after stimulation (Fig. 5i, j). Nevertheless, cells treated with a FAK inhibitor (FAKi) that effectively abolished FAK phosphorylation could still induce LRP6 phosphorylation upon Netrin-1 stimulation (Fig. 5k, l) demonstrating that Netrin-1 regulates LRP6 activation in brain ECs independently of FAK.

**Netrin-1 binding to Unc5B mediates BBB integrity.** To specifically interrogate whether blocking Netrin-1-Unc5B interactions disrupted the BBB in vivo, we used monoclonal antibodies (mAbs) that we had previously generated against the Unc5B IgG-like domains[29]. Anti-Unc5B-1 recognizes human but not mouse Unc5B, while anti-Unc5B-2 recognizes both human and mouse Unc5B and internalizes Unc5B[29]. Anti-Unc5B-2 treatment induced Unc5B internalization in brain ECs in vitro (Supplementary Fig. 4a) and i.v. injection of anti-Unc5B-2 for 1 h at 10 mg/kg in mice reduced brain Unc5B expression compared to anti-Unc5B-1 CTRL Ab-treated animals (Supplementary Fig. 4b,

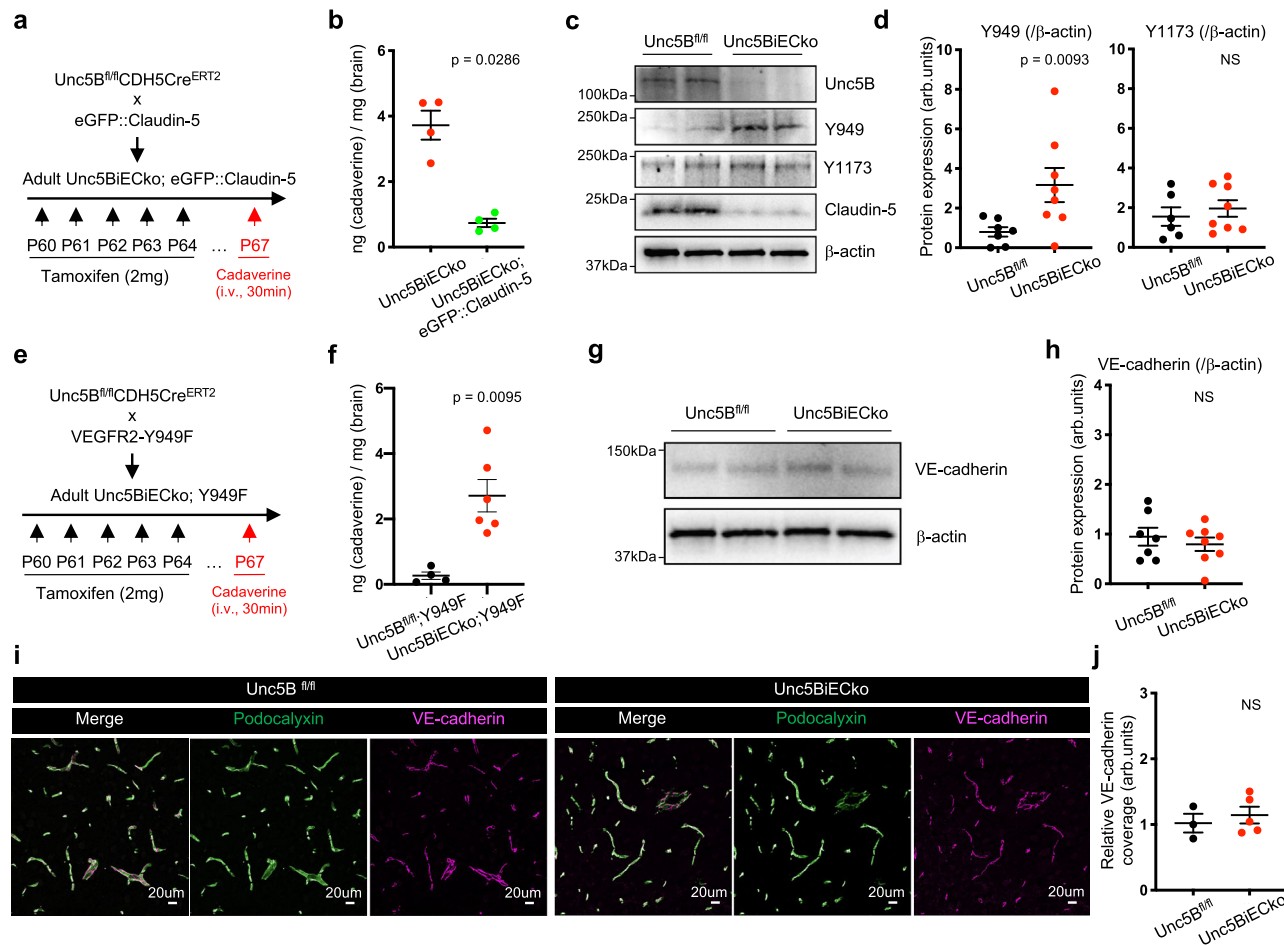

**Fig. 4 Claudin-5 but not VEGFR2 is involved in Unc5B BBB regulation. a** *Unc5B* deletion and *eGFP::Claudin*-5 gene overexpression strategy using tamoxifen injection. **b** Quantification of P67 brain cadaverine content, 30 min after i.v cadaverine injection, $n = 4$ *Unc5BiECko* and $n = 4$ *Unc5BiECko;eGFP::Claudin-5* brains. Each dot represents one mouse. Western blot (**c**) and quantification (**d**) of P67 brain protein extracts, $n = 7$ *Unc5B^{fl/fl}* and $n = 8$ *Unc5BiECko* brains. Each dot represents one mouse. One control mouse was set as 1. **e** *Unc5B* and *VEGFR2-Y949F* gene recombination strategy using tamoxifen injection. **f** Quantification of P67 brain cadaverine content, 30 min after i.v cadaverine injection, $n = 4$ *Unc5B^{fl/fl};Y949F* and $n = 6$ *Unc5BiECko;Y949F* brains. Each dot represents one mouse. Western blot (**g**) and quantification (**h**) of brain protein extracts, $n = 7$ *Unc5B^{fl/fl}* and $n = 8$ *Unc5BiECko* brains. Each dot represents one mouse. One control mouse was set as 1. Immunofluorescence with the indicated antibodies (**i**) and quantification (**j**) of VE-cadherin coverage on P67 brain sections. Each dot is the mean of several images, $n = 3$ *Unc5B^{fl/fl}* and $n = 5$ *Unc5BiECko* brains. One control mouse was set as 1. All data are shown as mean ± SEM. NS non-significant. Two-sided Mann–Whitney *U* test was performed for statistical analysis. Source data are provided as a Source Data file.

c). Internalization from the plasma membrane using this antibody is expected to prevent binding of all Unc5B ligands in vivo.

To generate a mAb that specifically blocked Netrin-1 binding without Unc5B internalization, we screened a human phage-derived library against the entire rat Unc5B ECD and identified anti-Unc5B-3, a mAb that bound both human and rat Unc5B with high affinity (Fig. 6a–c) but did not induce Unc5B internalization nor its degradation in vivo (Supplementary Fig. 4d–f). I.v. injection of anti-Unc5B-3 for 15 min at 10 mg/kg followed by cardiac perfusion and immunolabelling using an anti-human IgG antibody revealed anti-Unc5B-3 binding to brain arteries and capillaries of *Unc5B^{fl/fl}*, but no binding in the *Unc5BiECko* mice (Fig. 6d, e, Supplementary Fig. 5a), demonstrating specific binding of anti-Unc5B-3 to endothelial Unc5B. Anti-Unc5B-3 blocked Netrin-1-induced Src phosphorylation in brain ECs in vitro (Fig. 6f, g). To test effects on ligand binding in vivo, we injected control anti-Unc5B-1 or anti-Unc5B-3 i.v (10 mg/kg for 1 h), followed by Unc5B immunoprecipitation from brain lysates using a commercial antibody recognizing the Unc5B ECD. Western blotting revealed that anti-Unc5B-3

blocked Netrin-1 binding to Unc5B in vivo, while Robo4 and Flrt2 could still interact with Unc5B (Fig. 6h, i).

To test if antibody-mediated Unc5B blockade could induce BBB leak, we injected i.v. CTRL or anti-Unc5B antibodies for 1 h in adult WT C57BL/6J mice, followed by i.v. injection of cadaverine 30 min before sacrifice and analysis (Fig. 6j). In mice treated with CTRL anti-Unc5B-1 or IgG Ab, there were no signs of BBB disruption and injected cadaverine remained confined inside brain vessels (Fig. 6k, l). In contrast, mice treated with anti-Unc5B-3 or anti-Unc5B-2 for 1 h showed a significant leakage of injected cadaverine into the brain parenchyma (Fig. 6k, l), demonstrating that blocking Netrin-1 binding to Unc5B is sufficient to open the BBB. Interestingly, i.v. injection of CTRL anti-Unc5B-1, anti-Unc5B-2 and -3 for 8 h prior to cadaverine injection for 30 min did not induce any BBB leakage (Fig. 6m, n). Western blot against human IgG confirmed presence of all antibodies in the serum 1 h and 8 h after i.v. injection (Supplementary Fig. 4g), suggesting a transient BBB disrupting effect of anti-Unc5B-3 and -2 antibodies. The vascular barrier disrupting effect of anti-Unc5B-2 and -3 was specific to the brain,

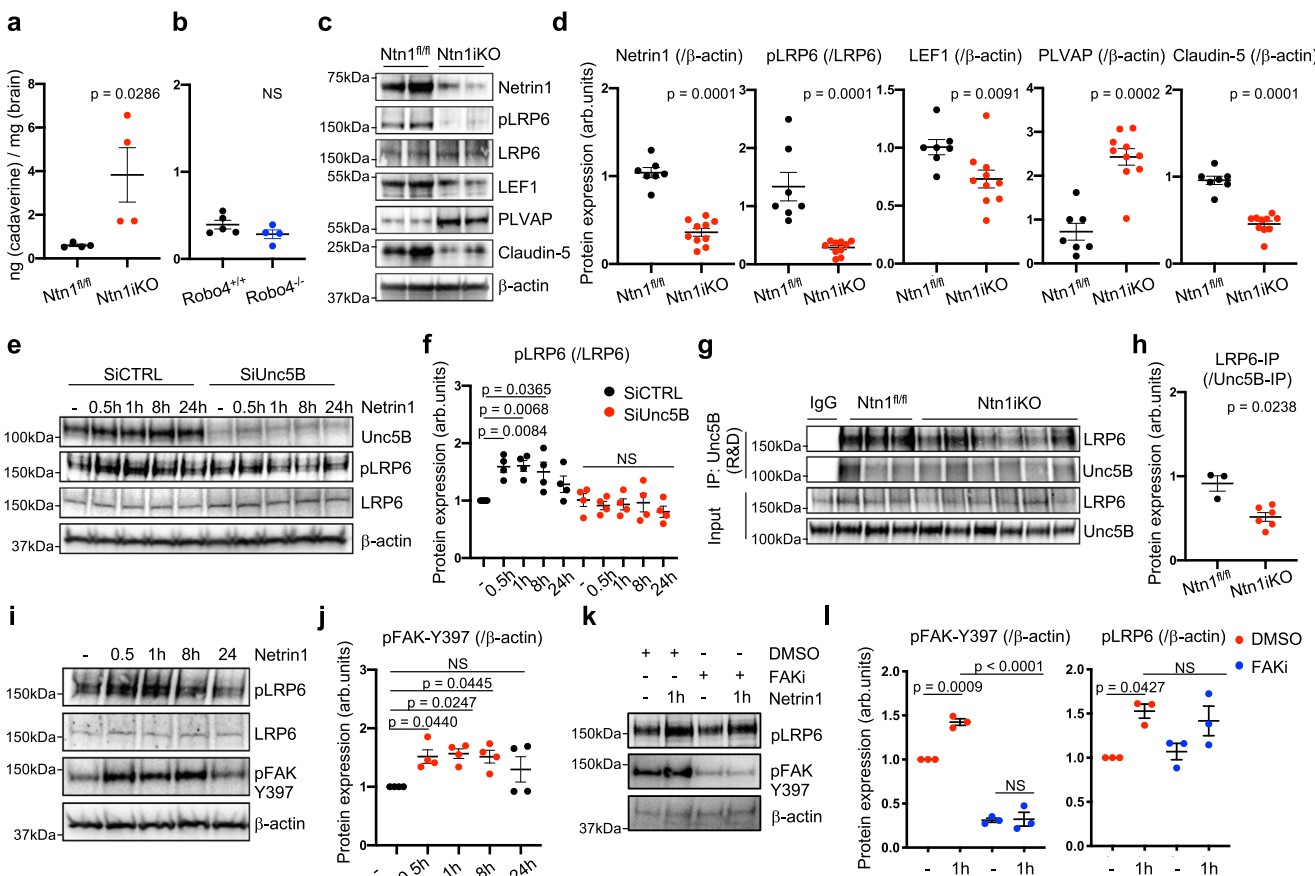

**Fig. 5 Netrin-1 controls BBB integrity. a**, **b** Quantification of cadaverine content in P67 brains, 30 min after i.v. cadaverine injection. *Ntn1* gene deletion was induced by tamoxifen injection between P60 and P64, *n* = 4 *Ntn1*^fl/fl^, *n* = 4 *Ntn1iko*, *n* = 5 *Robo4*^+/+^ and *n* = 4 *Robo4*^+/−^ brains. Each dot represents one mouse. Western blot (**c**) and quantification (**d**) of brain protein extracts, *n* = 7 *Ntn1*^fl/fl^ and *n* = 10 *Ntn1iko* brains. Each dot represents one mouse. One control mouse was set as 1. Western blot (**e**) and quantification (**f**) of mouse brain ECs treated with scrambled CTRL or Unc5B siRNA for 48 h and treated with recombinant mouse Netrin-1 (500 ng/ml) or not (−) for the indicated times. Each dot represents one independent experiment, *n* = 4 independent experiment. **g** CTRL IgG or Unc5B immunoprecipitation of brain protein extracts and Western blot for LRP6. **h** quantification of LRP6 pulldown with Unc5B, *n* = 3 *Ntn1*^fl/fl^ and *n* = 6 *Ntn1iko* brains. Each dot represents one mouse. One control mouse was set as 1. Western blot (**i**) and quantification (**j**) of mouse brain ECs treated with recombinant mouse Netrin-1 (500 ng/ml) or not (−) for the indicated times. Each dot represents one independent experiment, *n* = 4 independent experiment. Western blot (**k**) and quantification (**l**) of mouse brain ECs treated with CTRL DMSO or FAK inhibitor (5 uM) for 30 min followed by recombinant mouse Netrin-1 treatment (500 ng/ml) or not (−) for 1 h. Each dot represents one independent experiment, *n* = 3 independent experiment (pFAK-Y397(/β-actin): exact *p*-value between DMSO + Netrin1 (1 h) and FAKi + Netrin1 (1 h) = 0.000031). All data are shown as mean ± SEM. NS non-significant. Two-sided Mann–Whitney *U* test was performed for statistical analysis between two groups. ANOVA followed by Bonferroni's multiple comparisons test was performed for statistical analysis between multiple groups. Source data are provided as a Source Data file.

as tracer leakage in other organs 1 h after i.v. injection was similar between controls and anti-Unc5B-2 or -3 treated mice (Fig. 6o–q).

**Anti-Unc5B induced transient Wnt signaling inhibition.** To assess anti-Unc5B bioavailability and vascular clearance, we i.v. injected anti-Unc5B-3 antibodies (10 mg/kg) for 1 h, 8 h or 24 h followed by immunolabeling with anti-human IgG antibodies (Fig. 7a). Anti-Unc5B-3 was detectable in the brain vasculature 1 h after injection, declined to low levels after 8 h and was undetectable 24 h after injection (Fig. 7a–c), demonstrating that the antibody was unstable and rapidly cleared from the brain vasculature.

Interestingly, the expression of Wnt/β-catenin downstream targets varied in a similar time-dependent fashion. Claudin-5 immunostaining was downregulated 1 h after anti-Unc5B-3 injection and returned to basal levels after 8 h, whereas PLVAP immunostaining was upregulated at 1 and 8 h after anti-Unc5B-3 injection and returned to low baseline levels after 24 h (Fig. 7a).

Claudin-5 and PLVAP expression changes occurred in brain capillaries but not in larger vessels (>10 um, Fig. 7a, Supplementary Fig. 5b). Two-photon live imaging through cranial windows confirmed BBB leak from brain capillaries 1 h after anti-Unc5B-3 i.v. injection (Supplementary Fig. 5c, d).

Unc5B immunoprecipitation showed that anti-Unc5B-3 treatment transiently disrupted the Unc5B/LRP6 interaction 1 h after injection (Fig. 7b, c). Western blot on brain protein lysates from mice treated with CTRL IgG or anti-Unc5B-3 confirmed transient downregulation of Claudin-5 at 1 h, and upregulation of PLVAP at 1–8 h, and also showed transiently decreased LRP6 phosphorylation and decreased β-catenin protein levels 1–8 h after anti-Unc5B-3 injection that returned to baseline levels after 24 h (Fig. 7b, c).

**Size-selectivity of Unc5B mediated BBB opening.** To characterize the size selectivity of BBB opening induced by genetic or antibody-mediated Unc5B inhibition, we injected fluorescent dextrans of increasing molecular weights into mice. We observed that both 10 kDa and 40 kDa dextrans had a higher permeability across

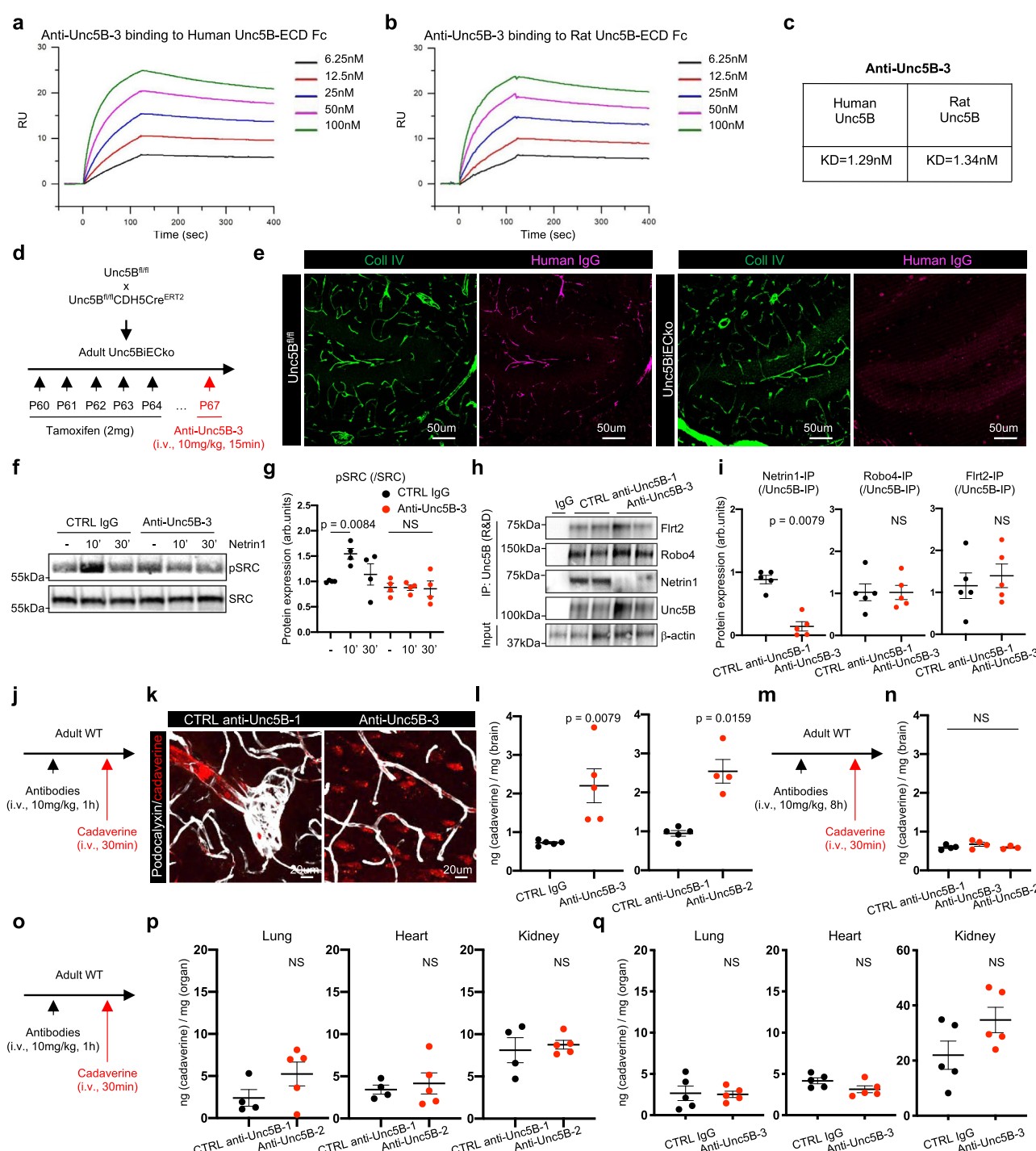

the BBB in *Unc5BiECko* brains and anti-Unc5B-2 and -3 treated animals compared to controls (Fig. 8a, b). By contrast, 70 kDa dextran, IgG and fibrinogen did not cross the BBB (Fig. 8c, Supplementary Fig. 6a, b), indicating a size-selective defect of BBB leakage for proteins greater than 40 kDa. As observed for cadaverine, leakage of dextrans in peripheral organs was similar between groups (Fig. 8d–f). Interestingly, leakage of 40 kDa dextran also remained increased in brains of *Unc5BiECko;eGFP::Claudin-5* mice compared to *Unc5BiECko* mice (Supplementary Fig. 6c). Besides dextrans, anti-Unc5B-3 treatment also enhanced delivery of single-chain nanobodies across the BBB when compared to CTRL IgG (Fig. 8g), while nanobody extravasation in other organs such as lung, heart, kidney or skin remained similar (Supplementary Fig. 6d). Moreover, injection of anti-Unc5B-3 enhanced brain delivery of BDNF and

induced phosphorylation and activation of its neuronal receptor Trk-B, while plasma BDNF levels remained similar to CTRL IgG injected mice (Fig. 8h–j), indicating that bioactive molecules up to 40 kDa can be delivered into the brain by this approach.

## Discussion

Our data reveal endothelial Unc5B as a critical regulator of BBB integrity. We showed that adult mice deficient in endothelial Unc5B expression exhibited widespread BBB leakage from brain capillary ECs, which converted from a Claudin-5+/PLVAP− BBB competent state to a leaky Claudin-5−/PLVAP+ state and displayed reduced expression of β-catenin and LEF1 (Supplementary Fig. 7). Combined heterozygous deletions of both Unc5B

**Fig. 6 Netrin-1 binding to Unc5B mediates BBB integrity. a**, **b** Surface Plasmon Resonance measurements of anti-Unc5B-3 binding to human and rat Unc5B-ECD-Fc. **c** Dissociation constant for anti-Unc5B-3 binding to human and rat Unc5B. **d** *Unc5B* gene deletion strategy using tamoxifen injection. **e** Anti-Unc5B-3 was i.v. injected in P67 *Unc5B^{fl/fl}* or *Unc5BiECko* mice for 15 min. Mice were perfused and anti-Unc5B-3 binding was detected by immunofluorescence on brain sections using an anti-human IgG antibody, reproduced on $n = 4$ *Unc5B^{fl/fl}* and $n = 3$ *Unc5BiECko* brain. Western-blot (**f**) and quantification (**g**) of ECs treated with CTRL IgG or anti-Unc5B-3 for 1 h followed by recombinant mouse Netrin-1 treatment (500 ng/ml) for 10 min or 30 min. Each dot represents one independent experiment, $n = 4$ independent experiment. **h** Unc5B immunoprecipitation with a commercial antibody (R&D systems) of brain protein extracts from mice i.v injected with CTRL or anti-Unc5B-3 antibodies (1 h, 10 mg/kg), and western blot with antibodies recognizing the indicated ligands. **i** Quantification of **h**, $n = 5$ CTRL anti-Unc5B-1 and $n = 5$ anti-Unc5B-3 treated animals. Each dot represents one mouse. One control mouse was set as 1. **j** I.v. antibody injection strategy. **k** Immunofluorescence staining on brain sections from antibody-injected mice. **l** Quantification of brain cadaverine content, $n = 5$ CTRL IgG, $n = 5$ anti-Unc5B-3, $n = 5$ CTRL anti-Unc5B-1 and $n = 4$ anti-Unc5B-2 treated animals. Each dot represents one mouse. **m** I.v. antibody injection strategy. **n** Quantification of brain cadaverine content, $n = 4$ CTRL anti-Unc5B-1, $n = 4$ anti-Unc5B-3 and $n = 3$ anti-Unc5B-2 treated animals. Each dot represents one mouse. **o** I.v. antibody injection strategy. **p**, **q** Quantification of cadaverine content in peripheral organs, $n = 4$ CTRL anti-Unc5B-1, $n = 5$ anti-Unc5B-2, $n = 5$ CTRL IgG and $n = 5$ anti-Unc5B-3 treated animals. Each dot represents one mouse. All data are shown as mean ± SEM. NS non-significant. Two-sided Mann–Whitney $U$ test was performed for statistical analysis between two groups. ANOVA followed by Bonferroni's multiple comparisons test was performed for statistical analysis between multiple groups. Source data are provided as a Source Data file.

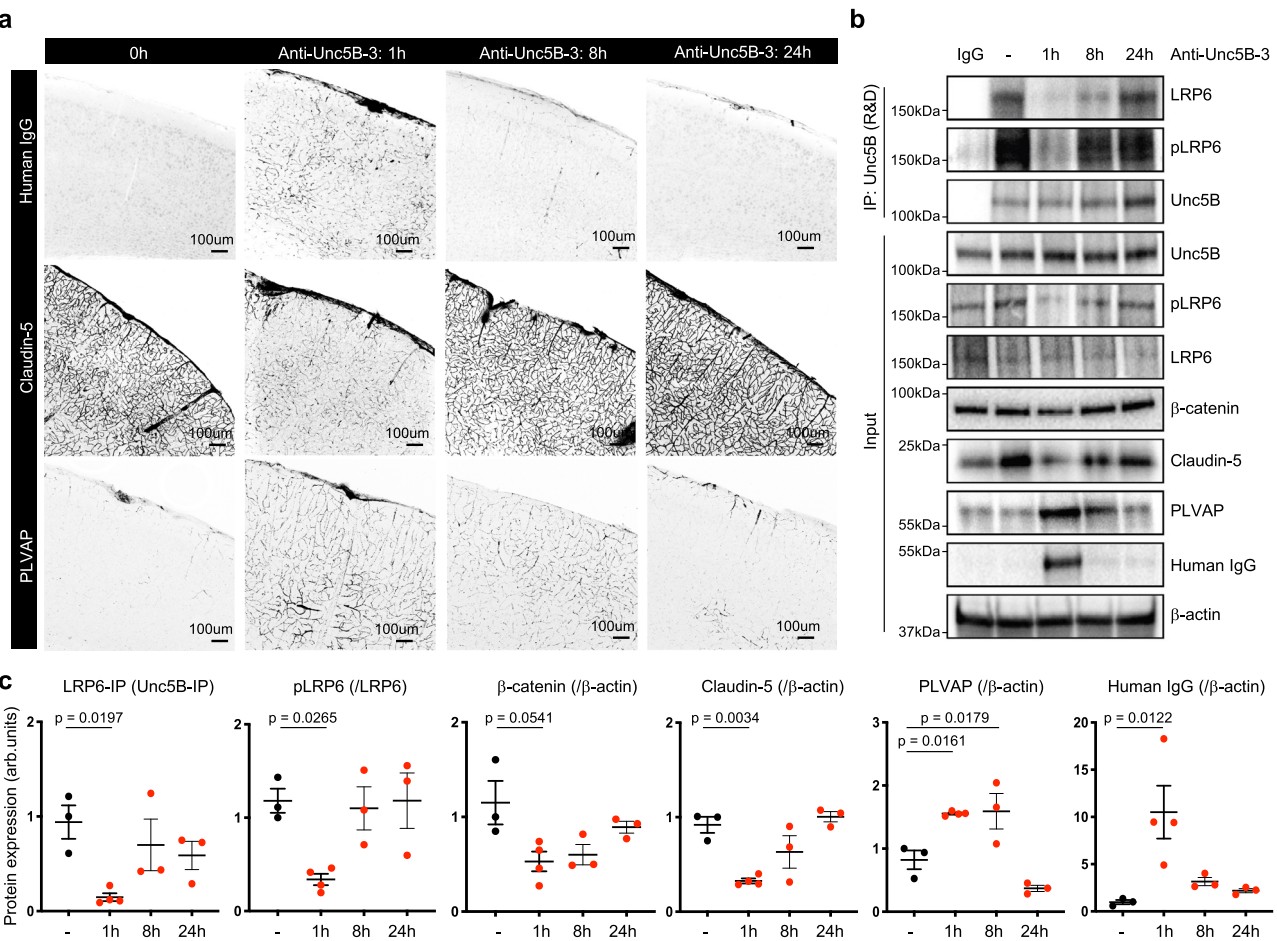

**Fig. 7 Wnt /β-catenin signaling regulation by Unc5B-blocking antibodies. a** Immunofluorescence staining of Human IgG, Claudin-5 and PLVAP and confocal imaging on brain sections from mice i.v. injected with anti-Unc5B-3 (10 mg/kg) for 1 h, 8 h or 24 h, reproduced on $n = 3$ untreated brains, $n = 4$ anti-Unc5B-3 treated brains for 1 h, $n = 3$ anti-Unc5B-3 treated brains for 8 and $n = 3$ anti-Unc5B-3 treated brains for 24 h. **b** CTRL IgG or Unc5B immunoprecipitation of brain protein extracts from mice i.v. injected with anti-Unc5B-3 and protein quantification (**c**), $n = 3$ control IgG treated brains, $n = 4$ anti-Unc5B-3 treated brains for 1 h and $n = 3$ anti-Unc5B-3 treated brains for 8 and $n = 3$ anti-Unc5B-3 treated brains for 24 h. Each dot represents one mouse. One control mouse was set as 1. All data are shown as mean ± SEM. NS non-significant. ANOVA followed by Bonferroni's multiple comparisons test was performed for statistical analysis between multiple groups. Source data are provided as a Source Data file.

and β-catenin induced BBB leak of cadaverine, while mice carrying single heterozygous deletions in either Unc5B or β-catenin displayed an intact BBB; and endothelial-specific β-catenin overexpression in *Unc5BiECko* mice increased Claudin-5 and LEF1 expression, while suppressing PLVAP and cadaverine leak,

together supporting that Unc5B maintains BBB integrity by functionally interacting with Wnt signaling.

Loss of endothelial Unc5B induced cadaverine leakage across the BBB in a widespread but non-uniform manner, in that caudal and ventral brain regions were more affected by loss of Unc5B

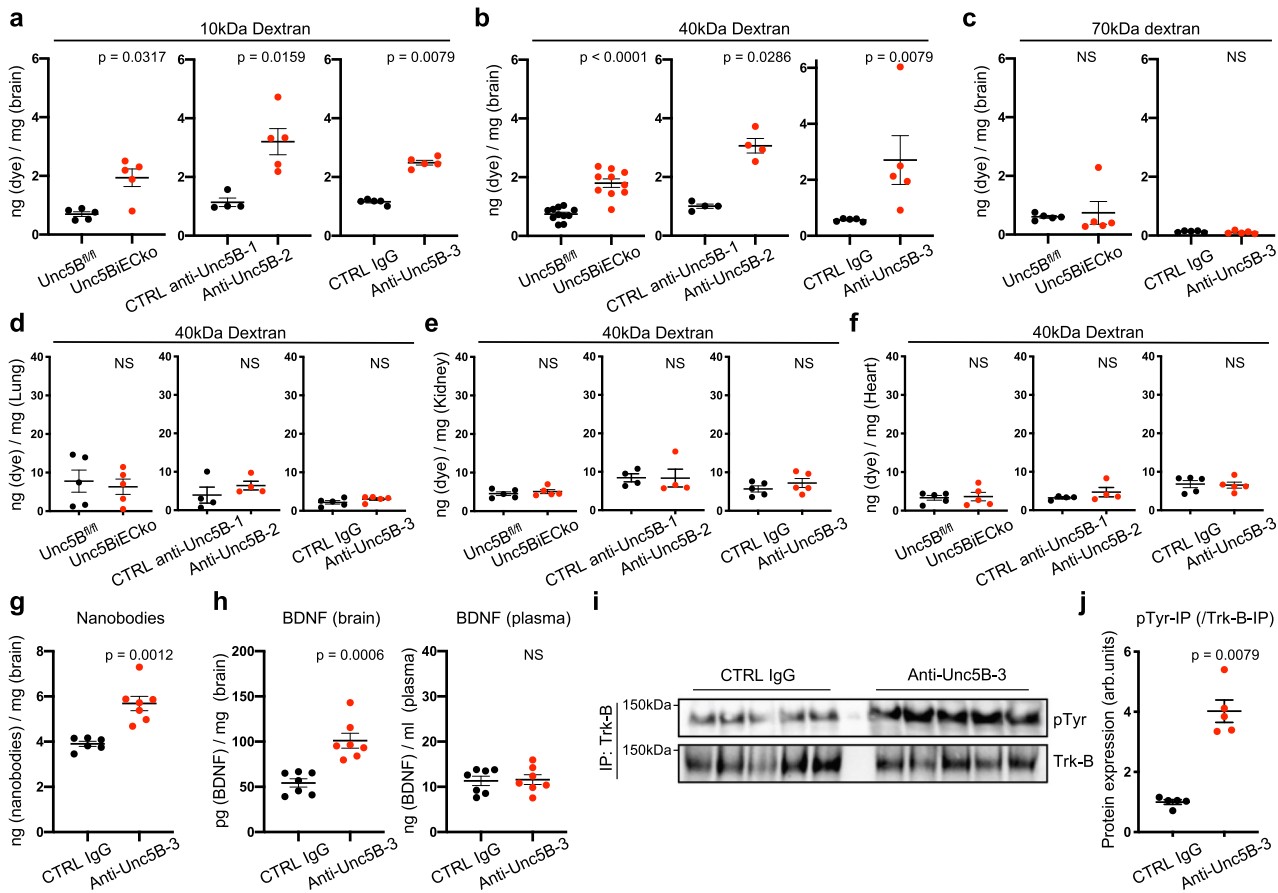

**Fig. 8 Size-selectivity of anti-Unc5B induced BBB opening.** Quantification of 10 kDa (**a**) 40 kDa (**b**) and 70 kDa (**c**) dextran content in P67 brains. Unc5B$^{fl/fl}$ and *Unc5BiECko* mice were injected with tamoxifen between P60-P64. CTRL anti-Unc5B-1, CTRL IgG, anti-Unc5B-2 and anti-Unc5B-3 antibodies were i.v. injected (10 mg/kg) for 1 h. Dextran was i.v. injected for 30 min. Quantification of 10 kDa dextran was performed on $n = 5$ *Unc5B$^{fl/fl}$* and $n = 5$ *Unc5BiECko* brains, $n = 4$ CTRL anti-Unc5B-1, $n = 5$ anti-Unc5B-2, $n = 5$ CTRL IgG and $n = 5$ anti-Unc5B-3 treated brains. Quantification of 40 kDa dextran was performed on $n = 11$ *Unc5B$^{fl/fl}$* and $n = 10$ *Unc5BiECko* brains (exact *p*-value = 0.000040), $n = 4$ CTRL anti-Unc5B-1, $n = 4$ anti-Unc5B-2, $n = 5$ CTRL IgG and $n = 5$ anti-Unc5B-3 treated brains. Quantification of 70 kDa dextran was performed on $n = 5$ *Unc5B$^{fl/fl}$* and $n = 5$ *Unc5BiECko* brains, $n = 5$ CTRL IgG and $n = 5$ anti-Unc5B-3 treated brains. Each dot represents one mouse. **d–f** Quantification of 40 kDa dextran content in P67 organs, $n = 5$ *Unc5B$^{fl/fl}$* and $n = 5$ *Unc5BiECko* brains, $n = 4$ CTRL anti-Unc5B-1, $n = 4$ anti-Unc5B-2, $n = 5$ CTRL IgG and $n = 5$ anti-Unc5B-3 treated brains. Each dot represents one mouse. **g** Quantification of P67 brain nanobody content 1 h after i.v CTRL IgG or anti-Unc5B-3 injection (10 mg/kg) and 30 min after i.v nanobody injection, $n = 6$ CTRL IgG and $n = 7$ anti-Unc5B-3 treated brains. Each dot represents one mouse. **h** Quantification of P67 brain and plasma BDNF concentration 1 h after i.v CTRL IgG or anti-Unc5B-3 injection (10 mg/kg) and 30 min in after i.v BDNF injection, $n = 7$ CTRL IgG and $n = 7$ anti-Unc5B-3 treated brains. Each dot represents one mouse. Trk-B immunoprecipitation of brain protein extracts from mice i.v. injected with CTRL IgG or anti-Unc5B-3 antibodies (10 mg/kg) for 1 h (**i**) and quantification of phospho-tyrosine pulldown (**j**), $n = 5$ CTRL IgG and $n = 5$ anti-Unc5B-3 treated brains. Each dot represents one mouse. One control mouse was set as 1. All data are shown as mean ± SEM. NS: non-significant. Two-sided Mann–Whitney *U* test was performed for statistical analysis between two groups. Source data are provided as a Source Data file.

than anterior and dorsal ones. This pattern of BBB disruption is roughly similar to the one observed in mice carrying inducible endothelial deletions of β-catenin[21,24]. We note that BBB leakage did not strictly correlate with Unc5B expression levels. Firstly, all cortical areas expressed endothelial Unc5B, but not all cortical areas were leaky in its absence. Secondly, Unc5B was detected in arteries and in capillaries, but only capillaries converted to a Claudin-5 negative, PLVAP-positive state in *Unc5BiECko* brains. The reasons for these region- and vessel segment-specific differences remain to be further investigated, but they may relate to β−catenin levels.

The specific Wnt ligands and receptors that maintain the BBB differ in a region-specific manner in the CNS, with cerebellum BBB utilizing Norrin signaling through LRP5, Fzd4 and TSPAN12, while cortex relies on Wnt7a/b signaling through LRP5/6, Fzd4, GPR124 and RECK[16,18–21,24]. Intercross between *Ndp* (Norrin) and *Wnt7a* mutants revealed that, with loss of

additional *Wnt7a* alleles in the *Ndp* KO background, there was a stepwise increase in the conversion of cerebellar ECs to a permeable state[21]. These and other genetic interaction studies between Wnt signaling components have suggested that a critical β-catenin threshold is required to maintain BBB ECs in a Claudin-5+/PLVAP− state[16,18,20,21,24]. Remarkably, blockade of Unc5B function affected both the cortex, cerebellum and other brain regions, suggesting that Unc5B may be an upstream regulator of both Wnt and Norrin signaling complexes at the BBB.

Mechanistically, we identify Netrin-1 as an Unc5B activating ligand required for BBB maintenance. Mice carrying global inducible deletions of the Unc5B ligand Netrin-1, and i.v. delivery of Unc5B mAbs that specifically block Netrin-1 binding to Unc5B (anti-Unc5B-3), or that induce receptor internalization (anti-Unc5B-2), all led to BBB breakdown and reduced Wnt/β-catenin signaling, while Robo4−/− mice displayed an intact BBB. These data support that Netrin-1 binding to Unc5B maintains BBB

integrity and Wnt signaling activation in CNS ECs. They confirm previous observations of Netrin's BBB-protective effects[52,53] and extend them in several ways. Podjaski et al.[52] showed that BBB junctional integrity was compromised in neonatal global *Ntn1* ko mice, however the neonatal lethality of these mice precluded analysis of the BBB at later developmental stages, and the relevant Netrin-1 receptor was not identified in that study. We extend these findings by showing that global inducible deletion of *Ntn1* in P60 mice induced BBB cadaverine leak, demonstrating that maintenance of adult BBB integrity requires Netrin-1. Furthermore, we identify the Unc5B receptor as the signal transducer on ECs, and we reveal canonical Wnt signaling as an effector pathway. Further experiments are required to determine the source of Netrin-1 mediating EC Unc5B signaling. Single cell RNA sequencing studies indicate that Netrin-1 is expressed in adult brain pericytes, fibroblasts, astrocytes and in ECs[49]. Conditional *Ntn1* deletion in astrocytes affects the BBB[53], and Netrin-1 is upregulated in ECs upon inflammatory signaling[52], therefore multiple cellular sources and environmental modulations of Netrin-1 expression could contribute to BBB integrity. Interestingly, serum Netrin-1 levels increased in patients with neuroinflammatory multiple sclerosis or type 2 diabetes[52,54]. Therefore, circulating Netrin-1 levels could be an important gatekeeper of BBB integrity, protecting the CNS and limiting BBB disruption during inflammatory conditions.

Mechanistically, the data show that Netrin-1 binding to Unc5B promotes LRP6 phosphorylation, and that the Unc5B intracellular UPA domain is required for complex formation between Unc5B and LRP6. Based on the crystal structure of the Unc5B ICD[33], we speculate that ligand-induced conformational changes in the Unc5B ICD expose the UPA domain and enable its interaction with LRP6. One possibility is that the UPA domain may induce LRP6 phosphorylation through recruitment of kinases. Recent studies in naïve pluripotent embryonic stem cells showed that Netrin-1 binding to Unc5B induced FAK-mediated phosphorylation of GSK3α/β, a kinase implicated in LRP6 activation[51], however our data show that Netrin-1 regulates LRP6 phosphorylation in brain ECs independently of FAK.

I.v. injected anti-Unc5B antibodies bound to the EC lumen in controls but not in *Unc5BiECko* brains, demonstrating Unc5B expression at the luminal side of brain ECs, which could be activated by Netrin-1 in the bloodstream. A critical question to be addressed in future work is how luminal Unc5B activates CNS Wnt signaling, which is believed to occur mainly at the abluminal side of CNS capillaries driven by neural progenitors or glia-derived WNTs and Norrin. Netrin-1-binding to luminal Unc5B could enhance Wnt/β-catenin signaling in endosomes or at cell-cell junctions, thereby facilitating subthreshold abluminal WNT BBB signaling effects.

Previous studies had speculated that transient Wnt signaling inhibition could be used to open the BBB "on-demand" for drug delivery into the diseased CNS[20,24], but the means to inhibit Wnt signaling in a CNS specific manner were not available. We demonstrated that antibody-mediated Unc5B blockade caused a transient loss of Wnt/β-catenin signaling and BBB breakdown for 1 h to 8 h followed by neurovascular barrier resealing, and allowed delivery of tracers up to 40 kDa into the adult CNS. The size selectivity of BBB opening is compatible with delivery of chemotherapeutics and of bioactive molecules such as nanobodies and growth factors in the specific regions of the brain where Unc5B regulates BBB integrity. Anti-Unc5B mAbs could therefore synergize with existing therapies such as focused ultrasound/microbubble approaches[55–58] and offer a therapeutic perspective for treatment of various human neurological disorders. Conversely, our data also raise the possibility that intravenous injection of Netrin-1 could increase Wnt/β-catenin signaling at

the BBB and repair CNS endothelial barrier breakdown in conditions such as multiple sclerosis or ischemic stroke where Netrin-1, Unc5B and several Wnt/β-catenin target genes are downregulated[59].

## Methods

**Mouse models.** All protocols and experimental procedures were approved by the Yale University Institutional Animal Care and Use Committee (IACUC). Animals were housed at 20–24 °C, with 30–70% humidity under a 12 h light-dark cycle. Generation of the targeted *Unc5b* allele was performed by homologous recombination in R1 ES cells. Correctly targeted cells were identified by Southern blot hybridization and injected into B6J blastocysts to generate *Unc5bneo/+* mice. To remove the neo cassette, *Unc5bneo/+* mice were mated to B6.129S4-Gt(ROSA) 26Sortm1(FLP1)Dym/RainJ mice (The Jackson Laboratory, stock #009086). Mice were backcrossed to B6J mice for ten generations. *Unc5B^{fl/fl}* (B6-Unc5b < tm1(flox) Slac/Slac) mice were then bred with *Cdh5CreERT2* mice[35] or *PDGFRβCreERT2*[37]. *TCF/LEF:H2B-GFP* mice, *eGFP::Claudin-5* mice, *Y949F* mice, βcatenin GOF *Ctnnb1^{flex/3}* mice, *Ctnnb1^{fl/fl}* mice, *Robo4^{−/−}* mice and *Ntn1^{fl/fl}* mice were described previously[29,39,43,44,46,48,50]. Gene deletion was induced by injection of tamoxifen (Sigma T5648) diluted in corn oil (Sigma C8267). Postnatal gene deletion was induced by 3 injections of 100ug of tamoxifen at P0, P1 and P2; whereas adult gene deletion was induced by 5 injections of 2 mg of tamoxifen from P60 to P64. Animals from both sexes were used.

**Cell culture.** C57BL/6 Mouse Primary Brain Microvascular Endothelial Cells were purchased from Cell Biologics (C57-6023). Cells were cultured in Dulbecco's Modified Eagle's Medium (DMEM) high glucose (Thermo Fisher Scientific, 11965092) supplemented with 10% fetal bovine serum (FBS) and 1% Penicillin Streptomycin at 37 °C and 5% CO$_2$ and split when confluent using Trypsin-EDTA (0.05%) (Life Technologies, 25300054). When indicated, cells were treated with recombinant Mouse Netrin-1 Protein (R&D, 1109-N1-025) at 500 ng/ml, or with 5uM of FAK inhibitor (GSK2256098, Selleckem S8523).

**Western-blot.** Brains were dissected and frozen in liquid nitrogen. They were lysed in RIPA buffer (Research products, R26200-250.0) supplemented with protease and phosphatase inhibitor cocktails (Roche, 11836170001 and 4906845001) using a Tissue-Lyser (5 times 5 min at 30 shakes/second). For western blot on cell culture, cells were washed with PBS and lysed in RIPA buffer with protease and phosphatase inhibitors cocktails. All protein lysates were then centrifuged 15 min at 16,100 × g at 4 °C and supernatants were isolated. Protein concentrations were quantified by BCA assay (Thermo Scientific, 23225) according to the manufacturer's instructions. 30 ug of protein were diluted in Laemmli buffer (Bio-Rad, 1610747) boiled at 95 °C for 5 min and loaded in 4–15% acrylamide gels (Bio-Rad, 5678084). After electrophoresis, proteins were transferred to a polyvinylidene difluoride (PVDF) membrane and incubated in TBS 0.1% Tween supplemented with 5% BSA for 1 h to block non-specific binding. The following antibodies were incubated overnight at 4 °C: Unc5B (Cell Signaling, 13851S, dilution: 1/1000), Robo4 (Invitrogen, 20221-1-AP, dilution: 1/300), Flrt2 (Novus bio, NBP2-43653, dilution: 1/500), Netrin-1 (R&D, AF1109, dilution: 1/500), Claudin-5 (Invitrogen, 35-2500, dilution: 1/1000), PLVAP (BD biosciences, 550563, dilution: 1/200), PDGFRβ (Cell Signaling, 3169S, dilution: 1/1000), GFAP (Millipore, MAB360, dilution: 1/1000), VEGFR2 Y949 (Cell Signaling, 4991S, dilution: 1/500), VEGFR2 Y1173 (Cell Signaling, 2478S, dilution: 1/500), pLRP6 (Cell Signaling, 2568S, dilution: 1/1000), LRP6 (Cell Signaling, 3395S, dilution: 1/500), βcatenin (Cell Signaling, 8480S, dilution: 1/2000), LEF1 (Cell Signaling, 2230S, dilution: 1/1000), ZO1 (Invitrogen, 61-7300, dilution: 1/2000), Occludin (Invitrogen, 33-1500, dilution: 1/1000), Caveolin-1 (Cell Signaling, 3238S, dilution: 1/2000), VE-cadherin (BD Pharmingen, 555289, dilution: 1/500) and βactin (Sigma, A1978, dilution: 1/5000). Then, membranes were washed 4x 10 min in TBS 0.1% Tween and incubated with one of the following peroxidase-conjugated secondary antibodies diluted 1/5000 in TBS 0.1% Tween supplemented with 5%BSA for 2 h at room temperature: horse anti-mouse IgG(H+L) (Vector laboratories, PI-2000), goat anti-rabbit IgG(H+L) (Vector laboratories, PI-1000), goat anti-rat IgG(H+L) (Vector laboratories, PI-9400), horse anti-goat IgG(H+L) (Vector laboratories, PI-9500). After 4x 10 min wash, western blot bands were acquired using ECL western blotting system (Thermo Scientific, 32106) or west femto maximum sensitivity substrate (Thermo Scientific, 34095) on a Biorad Gel Doc EQ System with Universal Hood II imaging system equipped with Image Lab software. See Supplementary Figs. 8–11 for the uncropped immunoblots.

**Immunoprecipitation.** Pierce™ protein A/G magnetic beads (Thermo fischer, 88802) were washed 5 times 10 min with RIPA buffer. 300 ug of protein lysate were diluted in 1 ml of RIPA buffer containing protease and phosphatase inhibitors and were incubated with 30 ul of A/G magnetic beads for 1 h at 4 °C under gentle rotation. Protein lysates were harvested using a magnetic separator (Invitrogen) and were incubated overnight at 4 °C under gentle rotation with 10 ug of Unc5B antibody (R&D, AF1006) or control IgG. The next day, 40 ul of protein A/G magnetic beads were added to each protein lysate for 2 h at 4 °C under gentle

rotation. Beads were then isolated using a magnetic separator and washed 5 times with RIPA buffer. After the last wash, supernatants were removed and beads were resuspended in 40ul of Laemmli buffer (Bio-Rad, 1610747), boiled at 95 °C for 5 min and loaded onto 4–15% gradient acrylamide gels. Western blotting was performed as described above.

**Immunostaining**. Brains were collected and placed in 3.7% formaldehyde overnight at 4 °C. Brains were then washed 3 times 10 min with TNT buffer (for 100 ml: 10 ml Tris 1 M pH7,4, 3 ml NaCl 5 M, 500 ul Triton X-100) and embedded in 2% agarose. 100 um sections were prepared using a Leica VT 1000S vibratome and placed in TNTB buffer (TNT buffer supplemented with 5% donkey serum) for 24 h at 4 °C. Primary antibodies were diluted in TNTB and placed for 48 h at 4 °C under gentle agitation. Then, sections were washed 5 times 30 min with TNT buffer and incubated for 24 h at 4 °C with secondary antibodies diluted in TNTB buffer. After 5x 30 min wash with TNT, sections were mounted using DAKO mounting medium (Agilent, S302380-2).

For mouse brain endothelial cell immunostaining, cells were seeded on 18 mm glass coverslips (Fischer Scientific, 12542A). When confluent, cells were washed with PBS and fixed with 3.7% formaldehyde for 10 min. Cells were washed 3x 5 min with PBS and were incubated with 0.2% TritonX100 diluted in PBS for an additional 10 min, washed 3 times and incubated with blocking solution (2%BSA, 3%Donkey serum diluted in PBS) for 1 h at room temperature. Primary antibodies were then diluted in blocking solution and incubated on coverslips overnight at 4 °C. After 3x 5 min washes, secondary antibodies diluted in blocking buffer were incubated on coverslips for 2 h at room temperature. Coverslips were then washed 3x 5 min with PBS and mounted using DAKO mounting medium.

The following antibodies were used: Podocalyxin (RD, AF1556, dilution: 1/400), Unc5B (Cell signaling, 13851S, dilution: 1/200), Claudin-5-GFP (Invitrogen, 352588, dilution: 1/200), PLVAP (BD biosciences, 550563, dilution: 1/100), LEF1 (Cell Signaling, 2230S, dilution: 1/200), GFAP (Millipore, MAB360, dilution: 1/400), Aquaporin-4 (Millipore, AB3068, dilution: 1/400), PDGFRβ (Cell Signaling, 3169S, dilution: 1/400), Endomucin (Hycult biotech, HM1108, dilution: 1/400), fibrinogen (DAKO, A0080, dilution: 1/400), CD13 (BD Biosciences, 558744, dilution: 1/200), VE-cadherin (BD Pharmingen, 555289, dilution: 1/200), DAPI (Thermo Fischer, 62248, dilution: 1/2000). All corresponding secondary antibodies were purchased from Invitrogen as donkey anti-primary antibody coupled with either Alexa Fluor 488, 568 or 647 and were diluted 1/200.

**Small interfering RNA knockdown experiments**. Cells were transiently transfected with siRNA (Dharmacon). ON-TARGETplus Mouse Unc5b siRNAs (SMARTpool, L-050737-01-0005) were used for Unc5B gene deletion. Transfection was performed using lipofectamine RNAi max (Invitrogen, 13778-075) according to the manufacturer's instruction with siRNA at a final concentration of 25pmol in OptiMem for 8 h. After transfection, cells were washed with PBS and fresh complete media was added for 48 h.

**Mouse lung endothelial cell isolation**. Mouse lungs were collected and minced into small pieces. Lungs were incubated in digestion buffer (5 ml of DMEM supplemented with 5 mg of collagenase I (Worthington LS004196), 10ul of 1 M Ca$^{2+}$ and 10ul of 1 M Mg$^{2+}$) for 1 h at 37 °C with shaking every 10 min. Once fully lysed, lung lysates were filtered through a 40um cell strainer (Falcon, 352340) into a solution of 3 ml FBS. Samples were centrifuged for 10 min at 200 g, and pellets were resuspended in PBS 0.1%BSA. In the meantime, rat anti-mouse CD31 (BD Pharmingen, 553370) was incubated with sheep anti-rat IgG magnetic dynabeads (Invitrogen, 11035) in a solution of sterile PBS 0.1%BSA (120ul of beads, 24 ul of antibodies in 12 ml PBS 0.1%BSA). Solutions were place under gentle rotation at room temperature for 2 h to allow proper coupling of antibodies and beads. Coupled beads were next isolated using a magnetic separator and incubated in the resuspended lung lysate for 30 min. After 5 washes with PBS 0.1%BSA, beads were separated using magnetic separator and seeded in 60 mm dishes containing mouse lung endothelial cell media (DMEM high glucose, 20%FBS, 1% Penicillin Streptomycin, 2% mitogen (Alta Aesar BT203). Purified endothelial cells were cultured at 37 °C and 5% CO$_2$ until confluence was reached, and then harvested.

**Quantitative real-time PCR analysis**. mRNA was isolated using Trizol reagent (Life Technologies, 15596018) according to the manufacturer's instructions and quantified RNA concentrations using nanodrop 2000 (Thermo Scientific). 300 ng of RNA were reverse transcribed into cDNA using iScript cDNA synthesis kit (Biorad, 170-8891). Real-time qPCR was then performed in duplicates using CFX-96 real-time PCR device (Bio-rad). Mouse GAPDH (QT01658692) was used as housekeeping gene for all reactions.

**Unc5B function-blocking antibody generation**. We performed a Phage-Fab (antigen-binding fragment) selection using a naïve Fab library (libF[60]) on a recombinant rat Unc5B-ECD Fc fusion protein (R&D systems) previously immobilized directly to Nunc Maxisorp plates starting at 5 ug/ml and dropping by 1 ug/ml per round. Phage titers from both input and output were monitored daily to ensure daily inputs were prepared correctly (minimum input of >1x 10E10) and output titers were at least 100-fold below input titers (day 1 output ~1x 10E3, day 2

output ~1x 10E4, day 3 output ~1x 10E5, days 4 and 5 output ~1x 10E7). Phage particles fused with Fabs were incubated with an unrelated protein (e.g., streptavidin) immobilized on a solid surface and allowed to bind in a step termed counterselection to remove unwanted phage-Fabs prior to incubation against target. After washing away unbound phage-Fab, phages were eluted from the target and amplified overnight for subsequent rounds. After 5 rounds of this process individual clones from rounds 3-5 were grown in 96-well format and tested by ELISA for their ability to bind antigen specifically. We selected several unique and different positive Fab over 5 rounds of selection, which were subcloned before antibody production (Proteogenix, Schiltigheim, France).

**Surface plasmon resonance**. Binding of anti-Unc5B antibodies to Human or Rat Unc5B was performed using a Biacore$^{TM}$ 8K (Proteogenix, Schiltigheim, France). Human or Rat Unc5B-ECD-Fc (R&D Systems) were immobilized on a CM5 sensor chip. Each antibody was diluted to gradient concentrations (50 nM, 25 nM, 12.5 nM, 6.25 nM, 3.125 nM) and flow through CM5 chip. The kinetic parameter was calculated using Bia-evaluation analysis software.

**Intravenous injection of antibodies, fluorescent tracer and nanobodies**. CTRL IgG, CTRL anti-Unc5B-1, anti-Unc5B-2 and -3 were injected intravenously into the lateral tail-vein of 8 weeks old adult mice at a concentration of 10 mg of antibodies/kg of mice and left to circulate from 1 h to 24 h depending on the experiment. All fluorescent tracers were injected intravenously into the lateral tail vein of 8 weeks old adult mice and left to circulate for 30 min. Lysine-fixable Cadaverine conjugated to Alexa Fluor-555 (Invitrogen) was injected at a concentration of 100 ug Cadaverine/20 g of mice. Lysine-fixable 10, 40 or 70 kDa dextran conjugated to tetramethylrhodamine (Invitrogen) were injected at a concentration of 250 ug dextran/20 g of mice. Nanobodies (Alexa Fluor-488 coupled anti-mouse nanobodies, Abnova) were injected at a concentration of 60 ug nanobodies/20 g of mice. For postnatal experiments, cadaverine was injected intraperitoneally into P5 neonates at a concentration of 250 ug cadaverine/20 g of pups and left to circulate for 2 h.

**Fluorescent tracer and nanobody quantification**. To assess tracer and nanobody leak, animals were perfused in the left ventricle with PBS. Brains (and other organs) were then collected, and their weight measured. Next, brains (and other organs) were incubated in formamide (Sigma-Aldrich, F7503) at 56 °C for 48 h. Dye fluorescence was then measured using a spectrophotometer at the appropriate emission and excitation wavelength. Dye extravasation from Unc5B$^{fl/fl}$, Unc5BiECko, WT treated with CTRL anti-Unc5B-1 and anti-Unc5B-2 were performed at the Yale Cardiovascular Research Center (New Haven, CT, USA) on a BioTek synergy2 spectrophotometer. Dye extravasation from WT treated with CTRL IgG2b and anti-Unc5B-3 were performed at the Paris Cardiovascular Research Center (Paris, France) on a Flexstation3 spectrophotometer. All results were normalized to the corresponding brain weight and reported to a standard made of known concentrations of dye diluted in formamide. Results are shown as ng of dye per mg of corresponding organ or tissue.

**BDNF extravasation quantification**. To assess BDNF leak, 1 h after antibody injection, 50 μg of Human BDNF diluted in saline was injected intravenously into the lateral tail vein in adult mice and left to circulate for 30 min. After sampling whole blood in EDTA-coated tubes, animals were perfused in the left ventricle with saline. Brains were then collected, and their weight measured. Blood was centrifuged at 1000 × g for 15 min at 4 °C, and plasma was then stored at −20 °C until use. Dissected brains were frozen in liquid nitrogen. They were lysed in RIPA buffer (Thermo Fisher) supplemented with protease and phosphatase inhibitor cocktails (Roche, 11836170001 and 4906845001) with increasing needle gauges and sonicated (3 times of 3 min each). All protein lysates were then centrifuged for 15 min at 14,000 × g at 4 °C and supernatants were isolated.

BDNF concentration in the plasma and brain lysates was quantified using a DuoSet BDNF ELISA Kit (R&D Systems, DY248) according to manufacturer instructions. Results were normalized to the corresponding brain weight. Results are shown as ng of nanobodies per mg of brain tissue.

**Two-photon live imaging**. Craniotomy was performed by drilling a 5-mm circle between lambdoid, sagittal, and coronal sutures of the skull on ketamine/xylazine anesthetized WT mice. After skull removal, the cortex was sealed with a glass coverslip cemented on top of the mouse skull. Live imaging was done 2 weeks later. For multiphoton excitation of endogenous fluorophores and injected dyes, we used a Leica SP8 DIVE in vivo imaging system equipped with 4tune spectral external hybrid detectors and an InSightX3 laser (SpectraPhysics). The microscope was equipped with in house designed mouse holding platform for intravital imaging (stereotactic frame, Narishige; gas anesthesia and body temperature monitoring/control, Minerve). Mice were injected intravenously with 10 mg/kg of CTRL IgG or anti-Unc5B-3 for 1 h followed by i.v injection of 10 kDa dextran for 1 h and image acquisition.

**Confocal microscopy and image analysis**. Confocal images were acquired on a laser scanning fluorescence microscope (Leica SP8 and Zeiss LSM800) using the appropriate software (LASX or ZEN system). 10X, 20X and 63X oil immersion objectives were used for acquisition using selective laser excitation (405, 488, 547, or 647 nm). Vascular density was quantified with the software Angiotool by quantifying the vascular surface area normalized to the total surface area. Quantification of pixel intensity was performed using the software ImageJ. VE-cadherin coverage analysis was performed on ImageJ by quantifying the VE-cadherin+ area normalized to the podocalyxin+ area. Vessel diameter was calculated on Leica LASX software.

**Statistical analysis**. All in vivo experiments were done on littermates with similar body weight per condition. Statistical analysis was performed using GraphPad Prism 8 software. Two-sided Mann–Whitney U test was performed for statistical analysis of two groups. ANOVA followed by Bonferroni's multiple comparisons test was performed for statistical analysis between 3 or more groups. Mantel-cox test was performed for survival curve statistical analysis.

**Reporting summary**. Further information on research design is available in the Nature Research Reporting Summary linked to this article.

## Data availability

All data generated are included in this article (main or supplementary information files) and are provided in the Source Data file. Additional information can be obtained from the corresponding author. Source data are provided with this paper.

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

## Acknowledgements

This project has received funding from the NIH (1R01HL149343-01 to A.E., 5R01NS35900 to S.L.A.), the Leducq Foundation (TNE ATTRACT, AE, LCW) and the European Research Council (ERC) (grant agreement No. 834161 to A.E.). K.B. was supported by a fellowship from the AHA (18POST34070109). We thank Max Thomas for technical assistance. S.L.A. is an investigator of the Howard Hughes Medical Institute.

## Author contributions

K.B., L.H.G., J.F., L.P.F., M.P., Y.X., L.J. and N.M. performed experiment. K.B. and A.E. designed experiments and analyzed data. A.E. and K.B. wrote the manuscript. D.K, B.N., D.A, L.C.W. and S.L.A., generated and provided essential reagents/mice.

## Competing interests

A.E., K.B., L.G. and L.P-F. are inventors on patent applications that cover the generation of Unc5B blocking antibodies, and their application. The remaining authors declare no competing interests.
