## [Peer Review File · Nature Communications]

Endothelial Unc5B controls blood-brain barrier integrityREVIEWER COMMENTS

Reviewer #1 (Remarks to the Author):

Boyé et al. report that Netrin1/Unc5B support the physiological level of Wnt/b-catenin signaling which is required in endothelial cells for the maintenance of BBB physiological homeostasis. They also observe that such function of Netrin1/Unc5B is specific for brain endothelial cells. In addition, only specific regions of the brain and endothelial cells of specific vessel kind show this type of BBB control.

General comments

These results are novel and add complexity to the picture of the regulation of the maintenance of BBB. However, several crucial aspects remain unfocused as detailed in 'Specific comments'. The authors also imagine therapeutic translation of their findings. This is speculative, as the manuscript is centred on physiological models of BBB, either established or dynamic, as in adult mice and in early pups. Indeed, the activity of such signaling in pathological conditions of the BBB is not investigated neither in mice nor in human brain samples.

In addition, the authors should clearly state at the end of the discussion that the potential therapeutic application of their finding is limited to the regions of the brain and to the type of endothelial cells in which Unc5B regulation of Wnt signaling is actually operating -as they describe-

.

Specific comments

- 1) The level of activation of b-catenin transcriptional signaling is mostly inferred, but not directly proofed. Tools directly reporting b-catenin transcriptional activity should be used in vitro and in vivo models to assess the basal state of Wnt/b-catenin signaling as well as the its modulation by Unc5B and Netrin1.
- 2) Working model in Fig S9. How is the binding of Unc5b to LRP6 increased upon Netrin binding?
- 3) Supp. Fig S6. Considering the specific increase of brain vessel permeability in the brain of adult mice, are Netrin1 and Unc5B expressed lung, heart and kidney of adult mice?
- 4) In models of increased BBB leakage is the level of Netrin and Unc5B down-regulated in parallel to decreased b-catenin signaling?
- 5) Fig 1c. and p10 lines 284-285- The authors report that specific brain regions show constant increase of permeability after Unc5B deletion. Are these brain areas also specifically enriched in Unc5B/Netrin1 expression?
- 6) Fig2 a-d. The authors propose a degradation mechanism to explain the reduction of b-catenin - and LEF1- after Unc5B deletion. However, direct measure of the transcripts of both b-catenin and LEF1 should be reported to clarify if Unc5B deletion affects protein stability only or also transcription.
- 7) p4 lanes 84-86. 'demonstrating that Unc5B has a CNS-selective BBB protective function in adult mice, which may be due to its enriched expression in adult brain endothelium when compared to endothelium of other organs.' Could the specific activity of Wnt in BBB maintenance also contribute to the specific activity of Unc5B in controlling brain vascular permeability?
In particular, Is Wnt signaling itself regulating the level of Unc5b and Netrin in a feed-back mechanism?
Are the expression levels of Unc5b and Netrin regulated in parallel to b-catenin/Wnt signaling during physiological BBB development? That is from high to low/maintenance level of b-catenin signaling?

8) Fig2e. Is LRP6 co-immunoprecipitated with Unc5B also phosphorylated? Is only phosphorylated LRP6 able to interact with Unc5B?
Phosphorylation of LRP6 should also be repeated in brain endothelial cells isolated from Unc5B ECKO mice besides the total brain extract shown in Fig 2a.

9) Fig 3n,o. Are claudin5 decreased and Plvap increased in the endothelial cells of brain vessels of mice treated with anti-Unc5B antibodies?

10) Fig2c. Lef1 seems to be expressed exclusively in podocalyxin positive cells (endothelial) is this specificity expected?

11) Fig 2f and g. The scheme and the WB should show the sequence of the Unc5B mutants in the same order.

12) Fig 3 e, f and respective comment p 7 line 182. 'This effect was abolished by Unc5B siRNA treatment (Fig.3 e,f)'. The 'NS' label on the figure 3f should be clarified, as it apparently is in contrast with the comment.

13) Fig 2i-n. Is b-catenin specifically deleted in endothelial cells? Is the activated b-catenin specifically expressed in endothelial cells? These aspects should be specified.

14) p6 line 158 and 159 and Fig 4S. 'Y949F mice compared to Cre-littermate controls (Supp. Fig. 4c,d), demonstrating that Vegfr2-Y949F failed to rescue BBB integrity in Unc5B mutant mice.'

What is therefore the role of increased Y949 VEGFR2 after Unc5B deletion? Is VE-cadherin phosphorylation increased in Unc5b deleted mice? Is VE-cadherin regularly distributed at endothelial cell-to-cell contacts in Unc5BECKO endothelial cells? (see also p7 lines 168-170).

15) Fig 3g,h. Phosphorylation of LRP6 co-immunoprecipitated with Unc5B should be tested directly, probing the co-IP with antibodies to phosphorylated LRP6. And point 13

16) p7 lines 184-186. 'suggesting that Netrin1binding to Unc5B regulated LRP6 phosphorylation and Wnt/b-catenin activation in CNS ECs.' How would Netrin-1 stimulate phosphorylation of LRP6?

17) (Supp. Fig.7a) and p.9 lines 232, 233. 'suggesting that Unc5B regulates BBB integrity mainly in arteries and capillaries'. Is permeability, measured as cadaverine or dextran 40kda leakage specifically increased in arteries and capillaries of the brain of Unc5BECKO mice?
In the same way, are claudin5 and Plvap specifically up-and down-regulated, respectively in in arteries and capillaries of the brain of Unc5BECKO mice?

18) Fig 4e,g,h. Is the acute effect of anti-Unc5B-3 antibody determined by destabilization/stabilization of Claudin5 and Plvap protein or by a transcriptional effect? rtPCR of Claudin5 and Plvap should be shown at different timepoints after anti-Unc5b-3 treatment.

19) Fig 4g. LRP6 phosphorylation decreased after anti-Unc5B-3 antibody?

20) Fig 4i,j and Fig S8. The authors should clarify why they used anti-Unc5B-2 instead of the most selective anti-Unc5B-3 for these in vivo experiments.

Reviewer #2 (Remarks to the Author):

In the present manuscript by Boyé et al. the authors have explored the function of Unc5B and its ligand Netrin1 for blood-brain barrier (BBB) function and maintenance. In endothelial-specific, inducible deletion mouse models, the authors could show that Unc5B deficiency in endothelial cells (EC) lead to BBB defects in a size-selective manner, with leakage of 1-40kDa traces but not of tracers larger than 70kDa. Moreover, due to the expression profile of Unc5B, mainly ventral areas of the brain as well as the cerebellum were affected whereas the dorsal cortical areas were essentially unaffected.

Mechanistically, Boyé et al. observed in the Unc5B^{IECKO} mice the Wnt/ β -catenin target Lef1 to be down regulated in ECs. Moreover, immunoprecipitation experiments revealed an interaction of the Unc5B cytoplasmic domain with the Wnt co-receptor Lrp6 in brain ECs, suggesting that Unc5B interacts with the Wnt/ β -catenin pathway, leading to a lack of BBB maintenance signal in Unc5B^{IECKO} mice. Given that Unc5B is a membrane receptor for Netrin1, Robo4 as well as for Flrt2, the authors clarified in systemic deletion mouse models that only Netrin1 deletion mimics the phenotype of the Unc5B^{IECKO} mice.

The authors could exclude that the BBB opening effects observed in the Unc5B^{IECKO} mice were mediated by other barrier/permeability-relevant pathways such as the VEGF/VEGFR2. However, Vegfr2-Y949F mutant mice in a Unc5B^{IECKO} background did not show any effect on permeability, suggesting that VEGF signaling may not directly be involved in BBB opening. Last but not least Boyé et al. explored the possibility to target Unc5B by antibodies to open the BBB for therapeutic intervention. By identifying antibodies that specifically target the Netrin1 binding domain of Unc5B, the authors could show on the one hand that Netrin1 binding is required for the BBB-maintaining function of Unc5B and that systemic antibody application can transiently open the BBB between 1-8hs.

The presented data are highly interesting to the field of brain vascular and BBB research but also to a broader audience, interested in regulation/modulation of Wnt/ β -catenin signaling in ECs. The work will likely contribute to the overall understanding of BBB regulation in health and potentially in disease, and might lead to the development of novel therapeutical strategies for the treatment of brain diseases with BBB dysfunction.

However, to augment the scientific merit of the manuscript, some issues require the authors attention:

Major points

The authors claim that the Netrin1/Unc5B signaling controls the passage of bio active molecules below 40kDa into the brain. What mechanism determines the cutoff at 40kDa? The authors should comment on paracellular vs. trans cellular routes and determine the trans cellular transport in KO mice.

The authors only explore the effect of Unc5B on BBB maintenance. What is the developmental expression profile of Unc5B and Netrin1 during the critical period of BBB induction when Wnt/ β -catenin signaling is at its peak (E13.5-15.5)? What is the effect of Unc5B deletion during embryonic brain angiogenesis?

At which membrane is Unc5B expressed, basolateral or apical/luminal? If systemic antibody application blocks Netrin1-Unc5B binding, is Netrin1 provided by the blood stream?

Along this line Frizzled/Lrp/Gpr124/Reck complex is mainly localized at the basolateral membrane. How does systemic Unc5B antibody application lead to effects on Lrp6 interaction? Could it be that Unc5B and Lrp6 interact independently of the Wnt receptor complex? Does Unc5B co-precipitate with Fzd4, Gpr124 and Reck?

Minor points

Page 3, line 43: Check the sentence „ Wnt/ β -catenin signaling maintains BBB integrity via expression of either Wnt7a,7b or Norrin ligands, which bind to multiprotein receptor...” this sounds somewhat wrong, as the sentence suggests that Wnt/ β -catenin signaling leads to the expression of Wnt7a/b and norrin and thereby drives its own signaling.

Page 3, line 48: Please correct throughout the manuscript “Claudin5” to “Claudin-5”. Also in the figures!!

Page 7, line 163: Change "BBB leakage of Cadaverine into..." into "BBB leakage of cadaverine into..."

Figure 2C: Add overlay or indicate the positive nuclei by asterisks or arrowheads. Moreover, there are some white spots in both LEF1 images in Fig. 2C which might be a left over from the pixel saturation tool of the confocal! Please check.

Figure 2M: Change in the graph headers "bactin" to " β -actin".

Figure 3N: What is the difference between the cadaverine+ nuclei highlighted by the yellow arrows, compared to the other cadaverine+ nuclei in the image?

Supp. Figure 2A: The cerebellum shows relatively weak Unc5B staining, comparable to the cortex. Still the cerebellum presented increased BBB leakage upon Unc5B KO in ECs. Please comment on this!

Supp. Figure 3B: Please correct "Aquaporin4" to "Aquaporin-4".

Overall the manuscript by Boyé et al. provides interesting and novel data on Unc5B at the BBB. Although some points raised above need to be addressed, the data are novel, solid and the presentation is of high quality. Hence the reviewer recommends the manuscript for publication after major revision.

Reviewer #3 (Remarks to the Author):

Boye et al report on studies examining the role of Netrin1 receptor Unc5B via conditional deletion using Cdh5CreERT2 mice. The authors convincingly demonstrate that endothelial Netrin1 receptor Unc5B maintains Wnt/b-catenin signaling, however, this is not a novel finding. The authors also show that intravenous delivery of antibodies blocking Netrin1 binding to Unc5B caused disruption of Wnt signaling and BBB breakdown, although studies examining the extent of this effect with regard to various vascular beds, and its complete kinetics appear preliminary. The authors also state that their purpose in generating Unc5Bx Cdh5CreERT2 mice is to specifically evaluate the role of Unc5B in brain endothelial cell function, especially barrier permeability (as the title states). However, as the authors mention, numerous tissues express both cadherin 5 and Unc5B, including the gastrointestinal (GI) tract, which expresses high levels of Unc5B protein. While the authors evaluated the effect of Unc5B deletion in several organs, the gut was not included. The paper cited by the authors (10.1038/s41593-019-0497-x) also did not include any GI tissues. Thus, it remains unclear if alterations in BBB permeability in Unc5Bfl/flCdh5Cre mice involve alterations in GI homeostasis, which have been shown to impact BBB expression of tight junction proteins. This needs to be addressed. Other major concerns follow:

Fig 1: Please confirm PLVAP upregulation in venular or capillary endothelial cells. Please also discuss the implications of this effect with regard to the role of PLVAP in endothelial cell function (aside from WNT signaling).

Fig 2: Please show the merged image for Podocalyxin and LEF1 immunostaining. The text calls out Fig 2n, which is not included; Fig 2j is likely the panel described. However, the genotype showing elevated cadaverine suggests that Unc5B is fl/wt, which would not lead to conditional deletion. Please explain.

Fig 3. Netrin1 has already been shown to regulate the Wnt/b-catenin pathway via Unc5B

(10.1038/s41556-020-0483-2). This study should be cited as it supports results in this figure; it also somewhat detracts from the novelty of the findings. The studies demonstrating a novel reagent for opening the BBB are interesting. Please demonstrate whether GI tract homeostasis is also impacted.

Fig 4. Arteries do not exhibit the specializations that constitute the BBB; these are exclusively found at post-capillary venules and capillaries. Please show whether arteries become more permeable in the setting of anti-Unc5B-3 i.v. injection. If so, these might lead to micro-hemorrhages. The MRI images are of poor resolution, making it difficult to assess the effect of anti-Unc5B-3. Please also include high quality images for all time points (can be supplemental). The MRI experiments may also not identify the peak time-point for BBB permeability, which may occur between 4 and 24 hours. Please provide additional time-points to address whether these findings are translatable.

Other concern

The manuscript contains concluding statements that are not supported by any data. For example, lines 64-67 state that seizures observed in neonatal mice with conditional deletion of Unc5B indicate "an abnormal excitability of the neuronal network that may result from a BBB failure." Although data provided indeed show BBB failure, no assessments of other tissues is provided, particularly from the gut, which could be contributing to altered brain homeostasis. A similar issue arises in stating that anti-Unc5B-1 CTRL Ab-treatment-mediated reduction in Unc5B "therefore preventing binding of all Unc5B ligands in vivo" without demonstrating this actually occurs. Please remove these statements

Reviewer #4 (Remarks to the Author):

This article describes the role for endothelial Unc5B in the BBB integrity. Authors performed well-organized and comprehensive experiments and provided robust and solid experimental results. The reviewer do not have major comments but some minor comments and questions as below.

"Unc5B function blocking antibody generation" in the method section: Authors performed five rounds of phage library screening, selected candidate clones, and tested them by ELISA for their ability to bind Unc5B-ECD Fc fusion protein specifically (page 16, lines 455-457). However, there was no description about how authors selected the anti-Unc5B-3 antibody that binds and blocks Unc5B but does not internalize it. What was the phage titer enrichment during the five rounds of screening? How was the rat Unc5B-ECD Fc fusion protein immobilized (e.g., coated on ELISA plates or protein A-beads)?

Page 9, lines 235-237: "Anti-Unc5B-3 was detectable in the brain vasculature 1h after injection, declined to low levels after 8h and was undetectable 24h after injection (Supp. Fig.7b), demonstrating rapid clearance from the brain vasculature." It is known that that antibodies generally circulate in the blood for three weeks. Why was the anti-Unc5B-3 antibody detected only for 24 h and rapidly cleared?

Page 3, lines 56-58: "Global Unc5B knockout in mice is embryonically lethal due to vascular defect." In this study, however, the blockade of Unc5B using antibody did not induce vascular leakage in organs (e.g., lung, heart, and kidney; Supp. Fig. 6) except brain. Why was the effect of anti-Unc5B-3 antibody specific to brain vasculature only? Please discuss about this.

Rebuttal letter

We thank the reviewers for the positive and constructive criticisms that we have addressed in full, as detailed below in a point-by-point response. Reviewers' comments are copied in bold font, our responses are in italic font below.

Reviewer #1 (Remarks to the Author):

Boyé et al. report that Netrin1/Unc5B support the physiological level of Wnt/b-catenin signaling which is required in endothelial cells for the maintenance of BBB physiological homeostasis. They also observe that such function of Netrin1/Unc5B is specific for brain endothelial cells. In addition, only specific regions of the brain and endothelial cells of specific vessel kind show this type of BBB control.

General comments

These results are novel and add complexity to the picture of the regulation of the maintenance of BBB.

We thank the reviewer for the positive comments on our manuscript.

However, several crucial aspects remain unfocused as detailed in 'Specific comments'. The authors also imagine therapeutic translation of their findings. This is speculative, as the manuscript is centred on physiological models of BBB, either established or dynamic, as in adult mice and in early pups. Indeed, the activity of such signaling in pathological conditions of the BBB is not investigated neither in mice nor in human brain samples. In addition, the authors should clearly state at the end of the discussion that the potential therapeutic application of their finding is limited to the regions of the brain and to the type of endothelial cells in which Unc5B regulation of Wnt signaling is actually operating -as they describe-.

→ To reflect this comment, we added in lines 400 to 403: "The size selectivity of BBB opening is compatible with delivery of chemotherapeutics and of bioactive molecules such as nanobodies and growth factors in the specific regions of the brain where Unc5B regulates BBB integrity."

Specific comments

1) The level of activation of b-catenin transcriptional signaling is mostly inferred, but not directly proofed. Tools directly reporting b-catenin transcriptional activity should be used in vitro and in vivo models to assess the basal state of Wnt/b-catenin signaling as well as its modulation by Unc5B and Netrin1.

→ We performed three sets of experiments to address this point. 1. qPCR analysis of global *Unc5B* KO brains at E12.5 revealed decreased *CLDN5* mRNA in *Unc5B* KO embryos compared to WT littermate controls, validating *Unc5B* deletion effects in an independent KO mouse strain (**revised Supp Fig. 2b**, revised text line 134 to 138). 2. qPCR on adult brain lysates revealed decreased mRNA levels of *LEF1* and *CLDN5* as well as increased *PLVAP* mRNA levels in *Unc5BiEcko* brains compared to Cre-negative littermate controls (**Revised Fig. 3a**). 3. we crossed *Unc5BiEcko* mice with *TCF/LEF:H2B-GFP* mice¹ which express a GFP reporter of β -catenin transcriptional activity. Compared to TAM-treated Cre-negative controls, *Unc5BiEcko;TCF/LEF:H2B-GFP* brains revealed decreased GFP expression in ERG+ ECs (**Revised Supp Fig. 3a-c**), attesting to decreased endothelial β -catenin transcriptional activity upon loss of *Unc5B* function. These results are described in the revised manuscript line 142 to 149.

2) Working model in Fig S9. How is the binding of Unc5b to LRP6 increased upon Netrin binding?

→ We clarify this in the introduction (line 66-73): *Unc5B* signaling is mediated by its intracellular domain (ICD), which encompasses a membrane-proximal ZU5 domain (named for its homology to ZO1), followed by a UPA domain (named for its conservation in *Unc5B*, *PIDD* and *Ankyrin*²) and a death domain (DD) that mediates apoptosis in the absence of ligand^{2,3}. These domains form a supramodule in which ZU5 binds to both UPA and DD suppressing *Unc5B* biological function, while ligand binding to *Unc5B* triggers a conformational change such that ZU5 loses its interaction with DD and exposes the UPA domain to activate *Unc5B* signaling².

In the discussion we write (line 379 to 386): Based on the crystal structure of the *Unc5B* ICD², we speculate that ligand-induced conformational changes in the *Unc5B* ICD expose the UPA domain and enable its interaction with LRP6. One possibility is that the UPA domain may induce LRP6 phosphorylation through recruitment of kinases. Recent studies in naïve pluripotent embryonic stem cells showed that Netrin1 binding to *Unc5B* induced FAK-mediated phosphorylation of *GSK3 α/β* , a kinase implicated in LRP6 activation⁴, however our data show that Netrin1 regulates LRP6 phosphorylation in brain ECs independently of FAK.

We experimentally tested whether FAK was implicated in LRP6 phosphorylation downstream of Netrin1 (line 231-237): We reasoned that Netrin1 could regulate LRP6

phosphorylation via FAK, a kinase that regulates β -catenin in pluripotent embryonic stem cells⁴. Netrin1-treated mouse brain ECs showed increased FAK phosphorylation from 1 to 8h after stimulation (**Fig. 5i,j**). Nevertheless, cells treated with a FAK inhibitor (FAKi) that effectively abolished FAK phosphorylation could still induce LRP6 phosphorylation upon Netrin1 stimulation (**Fig. 5k,l**) demonstrating that Netrin1 regulates LRP6 activation in brain ECs independently of FAK.

3) Supp. Fig S6. Considering the specific increase of brain vessel permeability in the brain of adult mice, are Netrin1 and Unc5B expressed lung, heart and kidney of adult mice?

→ We investigated *Unc5B* and *Netrin1* expression levels in organs using several RNAseq databases. Mouse brain ECs transcriptome data in health and disease models⁵ revealed enriched *Unc5B* expression in brain ECs compared to peripheral organs (**Rebuttal Fig. 1a**). *Netrin1* is also expressed by the brain vasculature⁵ (composed of ECs, fibroblasts and pericytes, **Rebuttal Fig. 1b**).

Rebuttal Fig. 1: (a,b) RNAseq analysis of *Unc5B* and *Ntn1* expression in adult brains (Munji et al, Nat Neuroscience 2019).

We discuss in line 368 to 376: Single cell RNA sequencing studies indicate that *Netrin1* is expressed in adult brain pericytes, fibroblasts, astrocytes and in ECs⁶. Conditional *Ntn1* deletion in astrocytes affects the BBB⁷, and *Netrin-1* is upregulated in ECs upon inflammatory signaling⁸, therefore multiple cellular sources and environmental modulations of *Netrin1* expression could contribute to BBB integrity. Interestingly, serum *Netrin1* levels increased in patients with neuroinflammatory Multiple sclerosis, EAE or type 2 diabetes^{8,9}. Therefore, circulating *Netrin1* levels could be an important gatekeeper of BBB integrity, protecting the CNS and limiting BBB disruption during inflammatory conditions.

4) In models of increased BBB leakage is the level of Netrin and Unc5B down-regulated in parallel to decreased b-catenin signaling?

→ We thank the reviewer for the interesting question. Munji *et al.*⁵ analyzed acute, subacute or chronic stroke models, known to have disrupted BBB. UNC5B and NTN1 along with LEF1 and CTNNB1 are significantly downregulated during acute and subacute stroke compared to control mice (**Rebuttal Fig. 2a-d**). Overall, this suggests that UNC5B and NTN1 levels are decreased in parallel with decreased Wnt/ β -catenin signaling in disease models where the BBB is disrupted.

Rebuttal Fig. 2: (a-d) RNAseq analysis of Unc5B, Ntn1, LEF1 and Ctnnb1 expression in the healthy or diseased adult brains.

5) Fig 1c. and p10 lines 284-285- The authors report that specific brain regions show constant increase of permeability after Unc5B deletion. Are these brain areas also specifically enriched in Unc5B/Netrin1 expression?

→ *Unc5B* expression in cortical ECs of wildtype mice was similar between areas that displayed more or less cadaverine leak in *Unc5BiEcko* brains (**revised Fig. 1g**), hence *Unc5B* expression alone was not sufficient to predict severity of BBB breakdown (revised manuscript line 108 to 111). We agree with the reviewer that the specific activity of *Unc5B* in controlling brain vascular permeability likely relates to its regulation of Wnt activity. We discuss region specificity as follows (lines 336-341): We note that BBB leakage did not strictly correlate with *Unc5B* expression levels. Firstly, all cortical areas expressed endothelial *Unc5B*, but not all cortical areas were leaky in its absence. Secondly, *Unc5B* was detected in arteries and in capillaries, but only capillaries converted to a Claudin-5 negative, PLVAP positive state in *Unc5BiEcko* brains. The reasons for these region- and vessel segment-specific differences remain to be further investigated, but they may relate to β -catenin levels.

6) Fig2 a-d. The authors propose a degradation mechanism to explain the reduction of b-catenin -and LEF1- after Unc5B deletion. However, direct measure of the

transcripts of both b-catenin and LEF1 should be reported to clarify if Unc5B deletion affects protein stability only or also transcription.

→ *This point was addressed above (see point 1)*

7) p4 lanes 84-86. 'demonstrating that Unc5B has a CNS-selective BBB protective function in adult mice, which may be due to its enriched expression in adult brain endothelium when compared to endothelium of other organs.' Could the specific activity of Wnt in BBB maintenance also contribute to the specific activity of Unc5B in controlling brain vascular permeability?

→ *This point was addressed above (point 5)*

In particular, Is Wnt signaling itself regulating the level of Unc5b and Netrin in a feed-back mechanism? Are the expression levels of Unc5b and Netrin regulated in parallel to b-catenin/Wnt signaling during physiological BBB development? That is from high to low/maintenance level of b-catenin signaling?

→ *To test whether Unc5B and Netrin1 expression are regulated by Wnt signaling via a feed-back mechanism, we performed WB on brain lysate from CTRL or endothelial β -catenin KO mice ($Ctnnb1^{fl/fl}CDH5Cre^{ERT2}$), as well as on from CTRL or β -catenin GOF mice ($Ctnnb1^{flex/3}CDH5Cre^{ERT2}$). We showed that loss or gain of endothelial β -catenin did not significantly change Unc5B or Netrin1 expression (**Revised Supp. Fig. 3d-i**), suggesting that Unc5B acted upstream of Wnt signaling. These results are described in the revised manuscript line 183 to 187.*

We believe that comparing Wnt and Unc5B/Netrin1 regulation between development and adulthood is outside the scope of this study, as the present work deals with mechanisms maintaining the adult BBB.

8) Fig2e. Is LRP6 co-immunoprecipitated with Unc5B also phosphorylated? Is only phosphorylated LRP6 able to interact with Unc5B?

Phosphorylation of LRP6 should also be repeated in brain endothelial cells isolated from Unc5B ECKO mice besides the total brain extract shown in Fig 2a.

→ *We performed immunoprecipitation of Unc5B on mouse brain lysates as well as on primary mouse brain ECs and show that LRP6 and pLRP6 were co-immunoprecipitated with Unc5B (**revised Fig. 3f and revised Fig. 7b**). In response to a reviewer 2 question, we also showed that immunoprecipitation of Unc5B pulled down other members of Wnt signaling receptor complexes including Fzd4 and GPR124*

(revised Fig. 3f). These results are described in the revised manuscript line 158 to 160, and line 294 to 299.

9) Fig 3n,o. Are claudin5 decreased and Plvap increased in the endothelial cells of brain vessels of mice treated with anti-Unc5B antibodies?

→ We show in revised Fig. 7a that Claudin-5 immunostaining was downregulated 1h after anti-Unc5B-3 injection and returned to basal levels after 8h, whereas PLVAP immunostaining was upregulated at 1 and 8h after anti-Unc5B-3 injection and returned to low baseline levels after 24h. This is described in the revised manuscript line 286 to 290.

10) Fig2c. Lef1 seems to be expressed exclusively in podocalyxin positive cells (endothelial) is this specificity expected?

→ Yes, this specificity is expected from brain single cell RNA seq.⁶ and from tabula muris¹⁰ (Rebuttal Fig. 3a,b).

Rebuttal Fig. 3 : (a,b) Single cell RNAseq analysis of LEF1 expression in adult brains from (a) Vanlandewijck, M., He, L. et al. Nature 2018 or (b) Tabula muris

11) Fig 2f and g. The scheme and the WB should show the sequence of the Unc5B mutants in the same order.

→ *The schematic was reordered and revised Fig. 3g now shows the same sequence as the western blot in revised Fig. 3h.*

12) Fig 3 e, f and respective comment p 7 line 182. ‘This effect was abolished by Unc5B siRNA treatment (Fig.3 e,f)’. The ‘NS’ label on the figure 3f should be clarified, as it apparently is in contrast with the comment.

→ *We apologize for the imprecise labeling. In revised Fig. 5e,f we changed every “non-stimulated” label by “-“ to indicate no treatment.*

13) Fig 2i-n. Is b-catenin specifically deleted in endothelial cells? Is the activated b-catenin specifically expressed in endothelial cells? These aspects should be specified.

→ *Again, we apologize if these aspects were not clear in the manuscript. We specified in the revised manuscript line 170 to 172 : “[...] we generated a TAM-inducible endothelial specific β -catenin deletion by crossing $Ctnnb1^{fl/fl}$ mice with $CDH5Cre^{ERT2}$ mice¹¹ (Fig. 3j, Supp. Fig. 3d-f)” and line 177 to 179 : “Next, we crossed $Unc5BiEcko$ with mice overexpressing a TAM-inducible activated form of β -catenin ($Ctnnb1^{flex/3}$ mice¹²), thereby enhancing endothelial Wnt/β -catenin signaling (Fig. 3l, Supp. Fig. 3g-i)”.*

14) p6 line 158 and 159 and Fig 4S. ‘Y949F mice compared to Cre-littermate controls (Supp. Fig. 4c,d), demonstrating that Vegfr2-Y949F failed to rescue BBB integrity in Unc5B mutant mice.’

What is therefore the role of increased Y949 VEGFR2 after Unc5B deletion? Is VE-cadherin phosphorylation increased in Unc5b deleted mice? Is VE-cadherin regularly distributed at endothelial cell-to-cell contacts in Unc5BECKO endothelial cells? (see also p7 lines 168-170).

→ *Thank you for the interesting question. Western blot analysis of $Unc5BiEcko$ brains and Cre-negative littermate controls revealed no difference in total VE-cadherin expression (Revised Fig. 4g,h). Furthermore, immunostaining of VE-cadherin in $Unc5BiEcko$ and Cre-negative littermate control brains revealed similar VE-cadherin expression and junctional staining between genotypes (revised Fig. 4i,j), attesting to the absence of adherens junction disassembly and degradation. This is described in the revised manuscript line 211 to 213.*

15) Fig 3g,h. Phosphorylation of LRP6 co-immunoprecipitated with Unc5B should be tested directly, probing the co-IP with antibodies to phosphorylated LRP6. And point 13

→ *To address this point, we performed immunoprecipitation of Unc5B on mouse brain lysates as well as on mouse brain ECs and show that LRP6 and pLRP6 were co-immunoprecipitated with Unc5B (revised Fig. 3f and revised Fig. 7b,c).*

16) p7 lines 184-186. 'suggesting that Netrin1 binding to Unc5B regulated LRP6 phosphorylation and Wnt/b-catenin activation in CNS ECs.' How would Netrin-1 stimulate phosphorylation of LRP6?

→ *We thank the reviewer for the relevant question. We reasoned that Netrin1 binding to Unc5B may regulate LRP6 phosphorylation through recruitment of kinases. In particular, Netrin1 could regulate LRP6 phosphorylation via FAK, a kinase activated by Netrin1 that regulates β -catenin in pluripotent embryonic stem cells⁴. Netrin1-treated mouse brain ECs showed increased FAK phosphorylation from 1 to 8h after stimulation (revised Fig. 5i,j). Nevertheless, cells treated with a FAK inhibitor (FAKi) that effectively abolished FAK phosphorylation could still induce LRP6 phosphorylation upon Netrin1 stimulation (revised Fig. 5k,l) demonstrating that Netrin1 regulates LRP6 activation in brain ECs independently of FAK. We described these results in the revised manuscript line 231 to 237 and discussed them line 379 to 386. See also response to point 2 above.*

17) (Supp. Fig.7a) and p.9 lines 232, 233. 'suggesting that Unc5B regulates BBB integrity mainly in arteries and capillaries'. Is permeability, measured as cadaverine or dextran 40kda leakage specifically increased in arteries and capillaries of the brain of Unc5BECKO mice? In the same way, are claudin5 and Plvap specifically up-and down-regulated, respectively in arteries and capillaries of the brain of Unc5BECKO mice?

→ *The reviewer raised an important point. When examining brain vessels for Claudin5 and PLVAP expression, we realized that only small vessels <10um in diameter converted to a Claudin-5 negative, PLVAP positive state in Unc5BiEcko and anti-Unc5B-3 treated brains, while larger vessels >10um did not (see revised Fig.2h, revised Supp. Fig. 2a for Unc5BiEcko brains; revised Fig. 7a and revised Supp. Fig. 5b for anti-Unc5B-3 treated brains). This suggests that BBB leakage in Unc5BiEcko brains likely originates from capillaries. We described the results in the manuscript line 131 to 134, and line 290 to 291.*

Furthermore, i.v. injection of anti-Unc5B-3 for 15min at 10mg/kg followed by cardiac perfusion and immunolabelling using an anti-human IgG antibody revealed anti-

Unc5B-3 binding to brain arteries and capillaries of $Unc5B^{fl/fl}$, but no binding in the $Unc5BiECKo$ mice (Revised Fig. 6d,e, Supp. Fig. 5a), and two-photon live imaging through cranial windows confirmed BBB leak from brain capillaries 1h after anti- $Unc5B-3$ i.v. injection (Revised Supp. Fig. 5c,d). We described these results line 253 to 256 and 292 to 293.

We discuss the results line 336 to 341: We note that BBB leakage did not strictly correlate with $Unc5B$ expression levels. Firstly, all cortical areas expressed endothelial $Unc5B$, but not all cortical areas were leaky in its absence. Secondly, $Unc5B$ was detected in arteries and in capillaries, but only capillaries converted to a Claudin-5 negative, PLVAP positive state in $Unc5BiECKo$ brains. The reasons for these region- and vessel segment-specific differences remain to be further investigated, but they may relate to β -catenin levels.

18) Fig 4e,g,h. Is the acute effect of anti- $Unc5B-3$ antibody determined by destabilization/stabilization of Claudin5 and Plvap protein or by a transcriptional effect? rtPCR of Claudin5 and Plvap should be shown at different timepoints after anti- $Unc5b-3$ treatment.

*→ We thank the reviewer for the pertinent comment. We showed in **revised Fig.7b,c** transiently decreased β -catenin protein levels 1h after i.v. anti- $Unc5B-3$ injection, suggesting transiently decreased β -catenin transcriptional activity, as seen in $Unc5BiECKo$ mice. However, we cannot formally exclude other potential mechanisms such as direct effects on destabilization or stabilization of protein levels by anti- $Unc5B$. We rephrased the discussion to reflect this as follows (line 322 to 331): We showed that adult mice deficient in endothelial $Unc5B$ expression exhibited widespread BBB leakage from brain capillary ECs, which converted from a Claudin5+/PLVAP- BBB competent state to a leaky Claudin5-/PLVAP+ state and displayed reduced expression of β -catenin and LEF1 (**Supp. Fig. 7**). Combined heterozygous deletions of both $Unc5B$ and β -catenin induced BBB leak of cadaverine, while mice carrying single heterozygous deletions in either $Unc5B$ or β -catenin displayed an intact BBB; and endothelial-specific β -catenin overexpression in $Unc5BiECKo$ mice increased Claudin5 and LEF1 expression, while suppressing PLVAP and cadaverine leak, together supporting that $Unc5B$ maintains BBB integrity by functionally interacting with Wnt signaling.*

19) Fig 4g. LRP6 phosphorylation decreased after anti- $Unc5B-3$ antibody?

*→ Western blot on brain protein lysates from mice treated with CTRL IgG or anti- $Unc5B-3$ confirmed transiently decreased phosphorylation of LRP6 1h to 8h after anti- $Unc5B-3$ injection that returned to baseline levels after 24h (**Revised Fig. 7b,c**). This is described in the revised manuscript line 295 to 299.*

20) Fig 4i,j and Fig S8. The authors should clarify why they used anti-Unc5B-2 instead of the most selective anti-Unc5B-3 for these in vivo experiments.

→ We replaced the two photon live imaging previously shown with anti-Unc5B-2 with one using anti-Unc5B-3 in **revised Supp. Fig.5c,d**.

We also removed the MRI experiments, based on a comment by reviewer 3. We show transient BBB opening following both anti-Unc5B-2 and anti-Unc5B-3 using cadaverine injections 1 and 8hrs after antibody treatment in **revised Fig. 6j-n**.

Reviewer #2 (Remarks to the Author):

In the present manuscript by Boyé et al. the authors have explored the function of Unc5B and its ligand Netrin1 for blood-brain barrier (BBB) function and maintenance. In endothelial-specific, inducible deletion mouse models, the authors could show that Unc5B deficiency in endothelial cells (EC) lead to BBB defects in a size-selective manner, with leakage of 1-40kDa traces but not of tracers larger than 70kDa. Moreover, due to the expression profile of Unc5B, mainly ventral areas of the brain as well as the cerebellum were affected whereas the dorsal cortical areas were essentially unaffected.

Mechanistically, Boyé et al. observed in the Unc5BiEcko mice the Wnt/ β -catenin target Lef1 to be down regulated in ECs. Moreover, immunoprecipitation experiments revealed an interaction of the Unc5B cytoplasmic domain with the Wnt co-receptor Lrp6 in brain ECs, suggesting that Unc5B interacts with the Wnt/ β -catenin pathway, leading to a lack of BBB maintenance signal in Unc5BiEcko mice. Given that Unc5B is a membrane receptor for Netrin1, Robo4 as well as for Flrt2, the authors clarified in systemic deletion mouse models that only Netrin1 deletion mimics the phenotype of the Unc5BiEcko mice.

The authors could exclude that the BBB opening effects observed in the Unc5BiEcko mice were mediated by other barrier/permeability-relevant pathways such as the VEGF/VEGFR2. However, Vegfr2-Y949F mutant mice in a Unc5BiEcko background did not show any effect on permeability, suggesting that VEGF signaling may not directly be involved in BBB opening. Last but not least Boyé et al. explored the possibility to target Unc5B by antibodies to open the BBB for therapeutic intervention. By identifying antibodies that specifically target the Netrin1 binding domain of Unc5B, the authors could show on the one hand that Netrin1 binding is required for the BBB-maintaining function of Unc5B and that systemic antibody application can transiently open the BBB between 1-8hs.

The presented data are highly interesting to the field of brain vascular and BBB research but also to a broader audience, interested in regulation/modulation of Wnt/ β -catenin signaling in ECs. The work will likely contribute to the overall understanding of BBB regulation in health and potentially in disease, and might lead to the development of novel therapeutical strategies for the treatment of brain diseases with BBB dysfunction.

We would like to thank the reviewer for the accurate summary and the positive comments on our manuscript.

However, to augment the scientific merit of the manuscript, some issues require the authors attention:

Major points

The authors claim the the Netrin1/Unc5B signaling controls the passage of bio active molecules below 40kDa into the brain. What mechanism determines the cutoff at 40kDa? The authors should comment on paracellular vs. trans cellular routes and determine the trans cellular transport in KO mice.

*→ Revised Fig.8a-c shows quantifications of dextran leakage across the BBB in Unc5BiECko and Anti-Unc5B-2 and -3 treated brains, revealing a cutoff between 40 and 70kDa dextran. The data establish that Unc5B blockade generates a size-selective BBB leakage, even though we cannot explain the cutoff fully at this moment in time. Unc5BiECko; eGFP::Claudin-5 mice which overexpress about 2-fold the normal amount of Claudin-5¹³ rescued cadaverine leak (**revised Fig. 4a,b**), but not leakage of 40kda dextran (**revised Supp. Fig. 6c**), which might involve the induction of PLVAP expression, a component of EC fenestrae and transcytotic vesicles¹⁴⁻¹⁷.*

We believe identifying the vascular permeability route in Unc5BiECko brains is very important, but that it will require extensive additional investigation, including a) transmission electron microscopy studies associated with dye injections and quantification of tracer-filled vesicles and junctions, b) analysis of different time points and brain regions (note that all our experiments were done at 7 days after TAM injection and 30 min dye perfusion). We are gearing up to do these experiments and we plan on comparing Unc5BiECko, Ntn1iko, Lrp5/6ko and Ctnn1iECko. We hope the reviewer agrees that this goes beyond the time frame of our study, and trust that we will follow up on this important question in future work.

The authors only explore the effect of Unc5B on BBB maintenance. What is the developmental expression profile of Unc5B and Netrin1 during the critical period

of BBB induction when Wnt/ β -catenin signaling is at its peak (E13.5-15.5)? What is the effect of Unc5B deletion during embryonic brain angiogenesis?

→ We believe that comparing Wnt and Unc5B/Netrin1 regulation between development and adulthood is outside the scope of this study, as the present work deals with mechanisms maintaining the adult BBB. However, we substantiate the effects of Unc5B deletion on Claudin5 and PLVAP expression in adult and in Unc5B^{-/-} embryonic brains (**revised Fig.2f-g, revised Supp. Fig.2b-e**). Unc5B affects developmental angiogenesis, as do Wnt signaling components, hence elucidating the pathway interplay in development will require extensive follow-up investigation that goes beyond the time frame of this revision.

At which membrane is Unc5B expressed, basolateral or apical/luminal? If systemic antibody application blocks Netrin1-Unc5B binding, is Netrin1 provided by the blood stream?

→ We thank the reviewer for yet another very pertinent question. Since we detect Unc5B antibodies bound to ECs after i.v. injection, luminal Unc5B expression plays a role in BBB regulation. We discuss sources of Netrin1 in the revised manuscript line 373 to 376: Interestingly, Netrin1 is detectable in mouse and human serum^{8,18}, and serum Netrin1 levels increased in patients with neuroinflammatory Multiple sclerosis, EAE or type 2 diabetes^{8,9}. Therefore, circulating Netrin1 levels could be an important gatekeeper of BBB integrity, protecting the CNS and limiting BBB disruption during inflammatory conditions. We also discuss the issue raised by the reviewer in lines 387-394: I.v. injected anti-Unc5B antibodies bound to the EC lumen in controls but not in Unc5B^{iE}CKo brains, demonstrating Unc5B expression at the luminal side of brain ECs, which could be activated by Netrin1 in the bloodstream. A critical question to be addressed in future work is how luminal Unc5B activates CNS Wnt signaling, which is believed to occur mainly at the abluminal side of CNS capillaries driven by neural progenitors or glia-derived WNTs and Norrin. Netrin1-binding to luminal Unc5B could enhance Wnt/ β -catenin signaling in endosomes or at cell-cell junctions, thereby facilitating subthreshold abluminal WNT BBB signaling effects.

To assess if circulating Netrin1 could regulate the Wnt signaling, we i.v. injected recombinant mouse Netrin1 in WT mice for 1h. Western blot on mouse brain lysate showed increased phosphorylation of LRP6 along with increased LEF1 protein level 1h after i.v. Netrin1 injection compared to PBS injected mice (**Rebuttal Fig. 4**), suggesting activation of the Wnt/ β -catenin signaling. We are following up on these findings by injecting Netrin1 into Unc5B^{iE}CKo mice, and by performing dose-dependence and kinetics of these results. We also generated inducible Ntn1 overexpressing mice and are testing BBB stabilizing effects in these mice. However, we hope the reviewer agrees that

completion of these studies requires extensive additional experimentation that we will perform in a follow-up study.

Rebuttal Fig. 4:
Western blot of brain protein extracts from mice injected with PBS or recombinant mouse Netrin1 (500 ug/kg) for 1h.

Along this line Frizzled/Lrp/Gpr124/Reck complex is mainly localized at the basolateral membrane. How does systemic Unc5B antibody application lead to effects on Lrp6 interaction? Could it be that Unc5B and Lrp6 interact independently of the Wnt receptor complex? Does Unc5B co-precipitate with Fzd4, Gpr124 and Reck?

→ Immunoprecipitation of *Unc5B* from primary microvascular mouse brain ECs pulled down LRP6, pLRP6, *Frizzled4* and the Wnt co-receptor GPR124 (**revised Fig. 3f**), demonstrating a physical interaction between *Unc5B*, LRP6, *Frizzled4* and GPR124 receptors. We also show in **revised Fig.7b-c** that anti-*Unc5B-3* treatment transiently blocks LRP6 phosphorylation in vivo, supporting interaction between *Unc5B* and LRP6 in vitro and in vivo at the BBB. These results are described in the revised manuscript line 158 to 160.

We also show below preliminary data supporting luminal localization of LRP6 in brain blood vessels (**Rebuttal Fig. 5**). We need to substantiate those in LRP6 ko mice to ascertain specificity, and are waiting for the mice to grow up to do so, but the data support our contention that *Unc5B* and LRP6 can interact at the luminal side of brain blood vessels and affect Wnt signaling.

Rebuttal Fig. 5: Antibodies recognizing the extracellular domain of LRP6 (LRP6-ECD), or the intracellular domain of LRP6 (LRP6-ICD) were i.v. injected for 30min. Mice were perfused and binding of LRP6 antibodies was detected by immunofluorescence staining.

Finally, analysis of human Genotype-Tissue Expression (GTEx) database revealed strong correlation between *Unc5B* expression and several Wnt coreceptors including *LRP6*, *ADGRA2* (encoding for *GPR124*) and *RECK* in brains, while this correlation was not detected in peripheral organs like in lungs. (**Rebuttal Fig. 6a,b**).

Rebuttal Fig. 6 : (a,b) Correlation analysis from the Genotype-Tissue Expression (GTEx) project

Minor points

Page 3, line 43: Check the sentence „ Wnt/ β -catenin signaling maintains BBB integrity via expression of either Wnt7a,7b or Norrin ligands, which bind to multiprotein receptor...” this sounds somewhat wrong, as the sentence suggests that Wnt/beta-catenin signaling Leads to the expression of Wnt7a/b and norrin and thereby drives its own signaling.

→ We changed the previous sentence to: “CNS specific Wnt7a,7b and Norrin ligands produced by glial cells bind to multiprotein receptor complexes including Frizzled4 and LRP6 on brain ECs”, line 52 to 54 of the revised manuscript.

Page 3, line 48: Please correct throughout the manuscript “Claudin5” to “Claudin-5”. Also in the figures!!

→ We corrected throughout the manuscript and figures “Claudin5” to “Claudin-5”

Page 7, line 163: Change “BBB leakage of Cadaverine into...” into “BBB leakage of cadaverine into...”.

→ We changed “BBB leakage of Cadaverine into...” into “BBB leakage of cadaverine into...” line 194 of the revised manuscript.

Figure 2C: Add overlay or indicate the positive nuclei by asterisks or arrowheads. Moreover, there are some white spots in both LEF1 images in Fig. 2C which might be a left over from the pixel saturation tool of the confocal! Please check.

→ Overlay images were added and pixel saturation were double checked.

Figure 2M: Change in the graph headers “bactin” to “β-actin”.

→ We changed graph headers “bactin” to “β-actin”.

Figure 3N: What is the difference between the cadaverine+ nuclei highlighted by the yellow arrows, compared to the other cadaverine+ nuclei in the image?

→ We apologized for the confusion. There is no difference between the cadaverine+ nuclei highlighted by the yellow arrows compared to the other cadaverine+ nuclei in the image. We removed the arrowheads in revised Fig. 1b and 6k.

Supp. Figure 2A: The cerebellum shows relatively weak Unc5B staining, comparable to the cortex. Still the cerebellum presented increased BBB leakage upon Unc5B KO in ECs. Please comment on this!

→ *Unc5B expression in cortical ECs of wildtype mice was similar between areas that displayed more or less cadaverine leak in Unc5BiEcko brains (revised Fig. 1g), hence Unc5B expression alone was not sufficient to predict severity of BBB breakdown (revised manuscript line 108 to 111). The specific activity of Unc5B in controlling brain vascular permeability may relates to its regulation of Wnt activity. We discuss region specificity as follows (lines 336-341): We note that BBB leakage did not strictly correlate with Unc5B expression levels. Firstly, all cortical areas expressed endothelial Unc5B, but not all cortical areas were leaky in its absence. Secondly, Unc5B was detected in arteries and in capillaries, but only capillaries converted to a Claudin-5 negative, PLVAP positive state in Unc5BiEcko brains. The reasons for these region- and vessel segment-specific differences remain to be further investigated, but they may relate to β-catenin levels.*

Supp. Figure 3B: Please correct “Aquaporin4” to “Aquaporin-4”.

→ We corrected “Aquaporin4” to “Aquaporin-4”.

Overall the manuscript by Boyé et al. provides interesting and novel data on Unc5B at the BBB.

Although some points raised above need to be addressed, the data are novel, solid and the presentation is of high quality. Hence the reviewer recommends the manuscript for publication after major revision.

Reviewer #3 (Remarks to the Author):

Boye et al report on studies examining the role of Netrin1 receptor Unc5B via conditional deletion using Cdh5CreERT2 mice. The authors convincingly demonstrate that endothelial Netrin1 receptor Unc5B maintains Wnt/b-catenin signaling, however, this is not a novel finding. The authors also show that intravenous delivery of antibodies blocking Netrin1 binding to Unc5B caused disruption of Wnt signaling and BBB breakdown, although studies examining the extent of this effect with regard to various vascular beds, and its complete kinetics appear preliminary. The authors also state that their purpose in generating Unc5Bx Cdh5CreERT2 mice is to specifically evaluate the role of Unc5B in brain endothelial cell function, especially barrier permeability (as the title states). However, as the authors mention, numerous tissues express both cadherin 5 and Unc5B, including the gastrointestinal (GI) tract, which expresses high levels of Unc5B protein. While the authors evaluated the effect of Unc5B deletion in several organs, the gut was not included. The paper cited by the authors (10.1038/s41593-019-0497-x) also did not include any GI tissues. Thus, it remains unclear if alterations in BBB permeability in Unc5B^{fl/fl}Cdh5Cre mice involve alterations in GI homeostasis, which have been shown to impact BBB expression of tight junction proteins. This needs to be addressed.

→ To address this issue, we analyzed permeability in brains and peripheral organs including GI tracts in TAM-injected Unc5B^{fl/fl}Cre mice and Cre-negative littermate controls. While loss of Unc5B induced increased cadaverine leak in the brain, no difference in dye permeability was seen in the GI tract of these animals (**Revised Fig. 1c**). Similarly, compared to TAM-treated Cre-negative controls, i.v. injection of 40kDa dextran induced leak in Unc5B^{fl/fl}Cre brains but not in their GI tracts (**Rebuttal Fig.7**). Therefore, endothelial Unc5B regulates BBB integrity independently of GI tract homeostasis.

Rebuttal Fig. 7: Quantification of dye content in brains and GI tracts 7 days after TAM injection and 30min after i.v. 40kDa dextran injection (n = 5 mice/group). All data are shown as mean+/-SEM. Mann-Whitney U test was performed for statistical analysis.

Other major concerns follow:

Fig 1: Please confirm PLVAP upregulation in venular or capillary endothelial cells. Please also discuss the implications of this effect with regard to the role of PLVAP in endothelial cell function (aside from WNT signaling).

→ When examining brain vessels for Claudin5 and PLVAP expression, we realized that only small vessels <10um in diameter converted to a Claudin-5 negative, PLVAP positive state in Unc5BiEcko and anti-Unc5B-3 treated brains, while larger vessels >10um did not (see **revised Fig.2h**, **revised Supp. Fig. 2a** for Unc5BiEcko brains; **revised Fig. 7a** and **revised Supp. Fig. 5b** for anti-Unc5B-3 treated brains). This suggests that BBB leakage in Unc5BiEcko brains likely originates from capillaries. We described the results in the manuscript line 131 to 134, and line 290 to 291.

Furthermore, i.v. injection of anti-Unc5B-3 for 15min at 10mg/kg followed by cardiac perfusion and immunolabelling using an anti-human IgG antibody revealed anti-Unc5B-3 binding to brain arteries and capillaries of Unc5B^{fl/fl}, but no binding in the Unc5BiEcko mice (**Revised Fig. 6d,e**, **Supp. Fig. 5a**), and two-photon live imaging through cranial windows confirmed BBB leak from brain capillaries 1h after anti-Unc5B-3 i.v. injection (**Revised Supp. Fig. 5c,d**). We described these results line 253 to 256 and 292 to 293.

We discuss the results line 336 to 341: We note that BBB leakage did not strictly correlate with Unc5B expression levels. Firstly, all cortical areas expressed endothelial Unc5B, but not all cortical areas were leaky in its absence. Secondly, Unc5B was detected in arteries and in capillaries, but only capillaries converted to a Claudin-5 negative, PLVAP positive state in Unc5BiEcko brains. The reasons for these region- and vessel segment-specific differences remain to be further investigated, but they may relate to β-catenin levels.

Fig 2: Please show the merged image for Podocalyxin and LEF1 immunostaining. The text calls out Fig 2n, which is not included; Fig 2j is likely the panel described. However, the genotype showing elevated cadaverine suggests that Unc5B is fl/wt, which would not lead to conditional deletion. Please explain.

→ *The merged image was placed in revised Fig.3d, and we apologize if the figure labeling was unclear.*

We are somewhat unclear about the comment regarding Fig.2n and j (now revised Fig. 3k and 3o), but we hope all the data and explanations are clear in the revised manuscript.

Fig 3. Netrin1 has already been shown to regulate the Wnt/b-catenin pathway via Unc5B (10.1038/s41556-020-0483-2). This study should be cited as it supports results in this figure; it also somewhat detracts from the novelty of the findings. The studies demonstrating a novel reagent for opening the BBB are interesting. Please demonstrate whether GI tract homeostasis is also impacted.

→ *The reference indicated by the reviewer (Huyghe et al. Nat Cell Biol 2020⁴) was already cited in the original manuscript line 334, and is now in the revised manuscript line 233 and line 384.*

*Huyghe et al.⁴ showed that Netrin signaling to Unc5B and Neogenin promotes naïve pluripotency in embryonic stem cells through Neo1 and Unc5B co-regulation of Wnt and MAPK signaling. Since Huyghe et al.⁴ showed that Netrin1 regulates β -catenin via FAK activation in pluripotent embryonic stem cells, we reasoned that Netrin1 could regulate LRP6 phosphorylation via FAK. We showed that Netrin1-treated mouse brain ECs increased FAK phosphorylation from 1 to 8h after stimulation (**Revised Fig. 5i,j**). Nevertheless, cells treated with a FAK inhibitor (FAKi) that effectively abolished FAK phosphorylation could still induce LRP6 phosphorylation upon Netrin1 stimulation (**Revised Fig. 5k,l**) demonstrating that Netrin1 regulates LRP6 activation in brain ECs independently of FAK. We described the results in the manuscript line 233 to 237 and discussed them line 377 to 386.*

Fig 4. Arteries do not exhibit the specializations that constitute the BBB; these are exclusively found at post-capillary venules and capillaries.

Please show whether arteries become more permeable in the setting of anti-Unc5B-3 i.v. injection. If so, these might lead to micro-hemorrhages.

→ *When examining brain vessels for Claudin5 and PLVAP expression, we realized that only small vessels <10um in diameter converted to a Claudin-5 negative, PLVAP positive state in Unc5BiEcko and anti-Unc5B-3 treated brains, while larger vessels*

>10um did not (see **revised Fig.2h**, **revised Supp. Fig. 2a** for *Unc5BiEcko* brains; **revised Fig. 7a** and **revised Supp. Fig. 5b** for anti-*Unc5B-3* treated brains). This suggests that BBB leakage in *Unc5BiEcko* brains likely originates from capillaries. We described the results in the manuscript line 131 to 134, and line 290 to 291 and discussed them line 338 to 340. Two-photon live imaging through cranial windows confirmed BBB leak from brain capillaries 1h after anti-*Unc5B-3* i.v. injection (**revised Supp. Fig. 5c,d**). We described these results line 292 to 293. Furthermore, we did not see any brain micro-hemorrhage, which is coherent with the size-selective limit of the BBB opening.

The MRI images are of poor resolution, making it difficult to assess the effect of anti-*Unc5B-3*. Please also include high quality images for all time points (can be supplemental). The MRI experiments may also not identify the peak time-point for BBB permeability, which may occur between 4 and 24 hours. Please provide additional time-points to address whether these findings are translatable.

→ We apologize for the poor MRI image resolution and agree with the reviewer that the peak time-point for BBB permeability might occur between 4 to 24h after anti-*Unc5B* injection.

Due to technical issues with the MRI facility we could not complete the requested additional MRI imaging over the time-period of this revision. We also note this would be a very expensive experiment. Therefore, we decided to remove the MRI data from the manuscript. Instead of MRI, we tested the reversibility of BBB opening at an intermediate time-point 8hrs. by i.v. injection of antibodies and quantification of cadaverine extravasation (**revised Fig. 6k-n**). In mice treated with CTRL anti-*Unc5B-1* or IgG Ab, there were no signs of BBB disruption and injected cadaverine remained confined inside brain vessels (**Fig. 6k,l**). In contrast, mice treated with anti-*Unc5B-3* or anti-*Unc5B-2* for 1h showed a significant leakage of injected cadaverine into the brain parenchyma (**Fig. 6k,l**), demonstrating that blocking Netrin1 binding to *Unc5B* is sufficient to open the BBB. Interestingly, i.v. injection of CTRL anti-*Unc5B-1*, anti-*Unc5B-2* and -3 for 8h prior to cadaverine injection for 30 minutes did not induce any BBB leakage (**Fig. 6m,n**). Western blot against human IgG confirmed presence of all antibodies in the serum 1h and 8h after i.v. injection (**Supp. Fig. 4g**), suggesting a transient BBB disrupting effect of anti-*Unc5B-3* and -2 antibodies. These results are described in the revised manuscript line 263 to 277.

Other concern

The manuscript contains concluding statements that are not supported by any data. For example, lines 64-67 state that seizures observed in neonatal mice with conditional deletion of *Unc5B* indicate “an abnormal excitability of the neuronal network that may result from a BBB failure.” Although data provided indeed show

BBB failure, no assessments of other tissues is provided, particularly from the gut, which could be contributing to altered brain homeostasis. A similar issue arises in stating that anti-Unc5B-1 CTRL Ab-treatment-mediated reduction in Unc5B “therefore preventing binding of all Unc5B ligands in vivo” without demonstrating this actually occurs. Please remove these statements

→ *We changed these statements line 84 to 86: “Interestingly, neonatal TAM injection induced seizures and lethality of Unc5B^{iE}Cko mice around P12 (Supp. Fig. 1d, Supp videos 1-4), indicating a possible BBB failure”, and line 247 to 248 : “Internalization from the plasma membrane using this antibody is expected to prevent binding of all Unc5B ligands in vivo.”.*

Reviewer #4 (Remarks to the Author):

This article describes the role for endothelial Unc5B in the BBB integrity. Authors performed well-organized and comprehensive experiments and provided robust and solid experimental results. The reviewer do not have major comments but some minor comments and questions as below.

→ *We would like to thank the reviewer for the critical and positive assessment of our manuscript.*

“Unc5B function blocking antibody generation” in the method section: Authors performed five rounds of phage library screening, selected candidate clones, and tested them by ELISA for their ability to bind Unc5B-ECD Fc fusion protein specifically (page 16, lines 455-457). However, there was no description about how authors selected the anti-Unc5B-3 antibody that binds and blocks Unc5B but does not internalize it. What was the phage titer enrichment during the five rounds of screening? How was the rat Unc5B-ECD Fc fusion protein immobilized (e.g., coated on ELISA plates or protein A-beads)?

→ *We revised the Unc5B function blocking antibody generation methods as follows. Rat Unc5B-ECD-Fc fusion protein was immobilized directly to Nunc Maxisorp plates starting at 5 ug/ml and dropping by 1 ug/ml per round. Phage titers from both input and output were monitored daily to ensure daily inputs were prepared correctly (minimum input of $> 1 \times 10^{10}$) and output titers were at least 100-fold below input titers (day 1 output $\sim 1 \times 10^3$, day 2 output $\sim 1 \times 10^4$, day 3 output $\sim 1 \times 10^5$, days 4 and 5 output $\sim 1 \times 10^7$). The method was completed in the revised manuscript line 546 to 560.*

We selected several unique binders and produced antibodies (Proteogenix, Schiltigheim, France) that were tested for their ability to bind Unc5B in vivo, induce Unc5B internalization, block Netrin-1 binding and open the BBB. We selected anti-Unc5B-3 as our lead candidate, as shown in revised Fig. 6a-n, revised Fig. 6q, revised Fig. 7a-c, Fig. 8a-j, Supp. Fig. 4d-g and Supp. Fig. 5a-d.

Page 9, lines 235-237: “Anti-Unc5B-3 was detectable in the brain vasculature 1h after injection, declined to low levels after 8h and was undetectable 24h after injection (Supp. Fig.7b), demonstrating rapid clearance from the brain vasculature.” It is known that that antibodies generally circulate in the blood for three weeks. Why was the anti-Unc5B-3 antibody detected only for 24 h and rapidly cleared?

→ We believe anti-Unc5B-3 could be quite unstable and subjected to fast degradation in vivo. Monoclonal antibody clearance from the vascular system depends on several parameters including injection routes, concentrations or targeted tissue^{19,20} and most antibodies going to the clinic require engineering to improve their stability, which was not the case for the anti-Unc5B antibodies.

*We tested the reversibility of BBB opening by i.v. injection of antibodies and quantification of cadaverine extravasation (**revised Fig. 6k-n**). In mice treated with CTRL anti-Unc5B-1 or IgG Ab, there were no signs of BBB disruption and injected cadaverine remained confined inside brain vessels (**Fig. 6k,l**). In contrast, mice treated with anti-Unc5B-3 or anti-Unc5B-2 for 1h showed a significant leakage of injected cadaverine into the brain parenchyma (**Fig. 6k,l**), demonstrating that blocking Netrin1 binding to Unc5B is sufficient to open the BBB. Interestingly, i.v. injection of CTRL anti-Unc5B-1, anti-Unc5B-2 and -3 for 8h prior to cadaverine injection for 30 minutes did not induce any BBB leakage (**Fig. 6m,n**). Western blot against human IgG confirmed presence of all antibodies in the serum 1h and 8h after i.v. injection (**Supp. Fig. 4g**), suggesting a transient BBB disrupting effect of anti-Unc5B-3 and -2 antibodies. These results are described in the revised manuscript line 263 to 277.*

Page 3, lines 56-58: “Global Unc5B knockout in mice is embryonically lethal due to vascular defect.” In this study, however, the blockade of Unc5B using antibody did not induce vascular leakage in organs (e.g., lung, heart, and kidney; Supp. Fig. 6) except brain. Why was the effect of anti-Unc5B-3 antibody specific to brain vasculature only? Please discuss about this.

→ We rephrased the Introduction to clarify that global homozygous Unc5B deletion is embryonically lethal at E12.5 due to placental vascular defects, as detailed in reference 31 (Tai-Nagara, I. et al. Development 2017²¹).

The specific activity of *Unc5B* in controlling brain vascular permeability likely relates to its regulation of Wnt activity. We discuss this as follows (lines 336-341): We note that BBB leakage did not strictly correlate with *Unc5B* expression levels. Firstly, all cortical areas expressed endothelial *Unc5B*, but not all cortical areas were leaky in its absence. Secondly, *Unc5B* was detected in arteries and in capillaries, but only capillaries converted to a Claudin-5 negative, PLVAP positive state in *Unc5BiEcko* brains. The reasons for these region- and vessel segment-specific differences remain to be further investigated, but they may relate to β -catenin levels.

References

- 1 Ferrer-Vaquero, A. *et al.* A sensitive and bright single-cell resolution live imaging reporter of Wnt/ss-catenin signaling in the mouse. *BMC Dev Biol* **10**, 121, doi:10.1186/1471-213X-10-121 (2010).
- 2 Wang, R. *et al.* Autoinhibition of UNC5b revealed by the cytoplasmic domain structure of the receptor. *Mol Cell* **33**, 692-703, doi:10.1016/j.molcel.2009.02.016 (2009).
- 3 Llambi, F., Causeret, F., Bloch-Gallego, E. & Mehlen, P. Netrin-1 acts as a survival factor via its receptors UNC5H and DCC. *EMBO J* **20**, 2715-2722, doi:10.1093/emboj/20.11.2715 (2001).
- 4 Huyghe, A. *et al.* Netrin-1 promotes naive pluripotency through Neol1 and Unc5b co-regulation of Wnt and MAPK signalling. *Nat Cell Biol* **22**, 389-400, doi:10.1038/s41556-020-0483-2 (2020).
- 5 Munji, R. N. *et al.* Profiling the mouse brain endothelial transcriptome in health and disease models reveals a core blood-brain barrier dysfunction module. *Nat Neurosci* **22**, 1892-1902, doi:10.1038/s41593-019-0497-x (2019).
- 6 Vanlandewijck, M. *et al.* A molecular atlas of cell types and zonation in the brain vasculature. *Nature* **554**, 475-480, doi:10.1038/nature25739 (2018).
- 7 Yao, L. L. *et al.* Astrocytic neogenin/netrin-1 pathway promotes blood vessel homeostasis and function in mouse cortex. *J Clin Invest* **130**, 6490-6509, doi:10.1172/JCI132372 (2020).
- 8 Podjaski, C. *et al.* Netrin 1 regulates blood-brain barrier function and neuroinflammation. *Brain* **138**, 1598-1612, doi:10.1093/brain/awv092 (2015).
- 9 Nedeva, I. *et al.* Relationship between circulating netrin-1 levels, obesity, prediabetes and newly diagnosed type 2 diabetes. *Arch Physiol Biochem*, 1-6, doi:10.1080/13813455.2020.1780453 (2020).
- 10 Tabula Muris, C. *et al.* Single-cell transcriptomics of 20 mouse organs creates a Tabula Muris. *Nature* **562**, 367-372, doi:10.1038/s41586-018-0590-4 (2018).
- 11 Brault, V. *et al.* Inactivation of the beta-catenin gene by Wnt1-Cre-mediated deletion results in dramatic brain malformation and failure of craniofacial development. *Development* **128**, 1253-1264 (2001).
- 12 Harada, N. *et al.* Intestinal polyposis in mice with a dominant stable mutation of the beta-catenin gene. *EMBO J* **18**, 5931-5942, doi:10.1093/emboj/18.21.5931 (1999).
- 13 Knowland, D. *et al.* Stepwise recruitment of transcellular and paracellular pathways underlies blood-brain barrier breakdown in stroke. *Neuron* **82**, 603-617, doi:10.1016/j.neuron.2014.03.003 (2014).

- 14 Clevers, H. Wnt/beta-catenin signaling in development and disease. *Cell* **127**, 469-480, doi:10.1016/j.cell.2006.10.018 (2006).
- 15 Liebner, S. *et al.* Wnt/beta-catenin signaling controls development of the blood-brain barrier. *J Cell Biol* **183**, 409-417, doi:10.1083/jcb.200806024 (2008).
- 16 Daneman, R. *et al.* Wnt/beta-catenin signaling is required for CNS, but not non-CNS, angiogenesis. *Proc Natl Acad Sci U S A* **106**, 641-646, doi:10.1073/pnas.0805165106 (2009).
- 17 Zhou, Y. *et al.* Canonical WNT signaling components in vascular development and barrier formation. *J Clin Invest* **124**, 3825-3846, doi:10.1172/JCI76431 (2014).
- 18 van Gils, J. M. *et al.* The neuroimmune guidance cue netrin-1 promotes atherosclerosis by inhibiting the emigration of macrophages from plaques. *Nat Immunol* **13**, 136-143, doi:10.1038/ni.2205 (2012).
- 19 Wang, W., Wang, E. Q. & Balthasar, J. P. Monoclonal antibody pharmacokinetics and pharmacodynamics. *Clin Pharmacol Ther* **84**, 548-558, doi:10.1038/clpt.2008.170 (2008).
- 20 Thomas, V. A. & Balthasar, J. P. Understanding Inter-Individual Variability in Monoclonal Antibody Disposition. *Antibodies (Basel)* **8**, doi:10.3390/antib8040056 (2019).
- 21 Tai-Nagara, I. *et al.* Placental labyrinth formation in mice requires endothelial FLRT2/UNC5B signaling. *Development* **144**, 2392-2401, doi:10.1242/dev.149757 (2017).

REVIEWERS' COMMENTS

Reviewer #1 (Remarks to the Author):

The authors have performed considerable experimental work to clarify the aspects that were questionable and obscure in the previous version of the manuscript. They adequately present and discuss these data in their revised manuscript.

Reviewer #2 (Remarks to the Author):

In the revised manuscript by Boyé et al. the authors have addressed most of the points raised by the reviewer and have considerably improved the manuscript scientifically as well as formally. Although the authors argue that the detailed exploration of the size-selective barrier opening in Unc5BiEcko mice is beyond the scope of the present study, which is reasonable, the revised manuscript provides data to support a paracellular leakage of cadaverine and a transcellular leakage of >40kDa dextran.

Similarly, the detailed developmental analysis of Unc5B and Wnt/ β -catenin interaction requires an additional study, the authors provide initial evidence for a this signaling axis in the embryonic brain.

All other major points have been fully addressed.

The manuscript in its current form is acceptable for publication and will certainly contribute to the overall understanding of BBB regulation in health and potentially in disease.

Some minor issues should however be addressed by the authors:

Minor points

Page 6, line 132: Change the sentence „Claudin-5 negative, PLVAP positive state occurred...” to „Claudin-5-negative, PLVAP-positive state occurred...”.

Page 11, line 290: Please correct “Claudin5 and PLVAP expression...” to “Claudin-5 and PLVAP expression...”.

Page 12, line 324: Change “Claudin5+/PLVAP- BBB competent state to a leaky Claudin5-/PLVAP+ state...” to “Claudin-5+/PLVAP- BBB competent state to a leaky Claudin-5-/PLVAP+ state...”.

Page 12, line 329: Change “...increased Claudin5 and LEF1 expression,...” to “...increased Claudin-5 and LEF1 expression,...”.

Page 13, line 373: Check the following sentence „Interestingly, serum Netrin-1 levels increased in patients with neuroinflammatory Multiple sclerosis, EAE or type 2 diabetes^{52,54}.” The syntax suggests that „EAE” is also a pathological condition in human.

Page 16, line 456: Change „..., Caveolin1 (Cell Signaling, 3238S), VE-cadherin...” to „..., Caveolin-1 (Cell Signaling, 3238S), VE-cadherin...”.

Suppl Figure 2A: Change „Claudin5” to „Claudin-5”.

Suppl Figure 5B: Change „Claudin5” to „Claudin-5”.

Figure legend Suppl Figure 5: Change „Quantification of Claudin5 and PLVAP immunostaining...” to „Quantification of Claudin-5 and PLVAP immunostaining...”.

Suppl Figure 8A: Change „Caveolin1“ to „Caveolin-1“, and change „ZO1“ to „ZO-1“.

Reviewer #3 (Remarks to the Author):

The authors have addressed my concern with text changes, additional data and removal of unsubstantiated claims and limited data sets. The manuscript is markedly improved by the revision.

Reviewer #4 (Remarks to the Author):

The authors addressed most of the comments raised by the reviewer. I do not have further comments or questions.

REBUTTAL LETTER

We thank the reviewers for the positive comments. Below is a point-by-point response to the remaining minor points. Reviewers' comments are copied in bold font, our responses are in italic font below.

Reviewer #1 (Remarks to the Author):

The authors have performed considerable experimental work to clarify the aspects that were questionable and obscure in the previous version of the manuscript. They adequately present and discuss these data in their revised manuscript.

→ We thank the reviewer for the critical assessment of our manuscript.

Reviewer #2 (Remarks to the Author):

In the revised manuscript by Boyé et al. the authors have addressed most of the points raised by the reviewer and have considerably improved the manuscript scientifically as well as formally.

Although the authors argue that the detailed exploration of the size-selective barrier opening in Unc5BiEcko mice is beyond the scope of the present study, which is reasonable, the revised manuscript provides data to support a paracellular leakage of cadaverine and a transcellular leakage of >40kDa dextran. Similarly, the detailed developmental analysis of Unc5B and Wnt/ β -catenin interaction requires an additional study, the authors provide initial evidence for a this signaling axis in the embryonic brain.

All other major points have been fully addressed.

The manuscript in its current form is acceptable for publication and will certainly contribute to the overall understanding of BBB regulation in health and potentially in disease.

Some minor issues should however be addressed by the authors:

Minor points

Page 6, line 132: Change the sentence „Claudin-5 negative, PLVAP positive state occurred...“ to „Claudin-5-negative, PLVAP-positive state occurred...“.

→ We changed “Claudin-5 negative, PLVAP positive state occurred...” to “Claudin-5-negative, PLVAP-positive state occurred...”.

Page 11, line 290: Please correct “Claudin5 and PLVAP expression...” to “Claudin-5 and PLVAP expression...”.

→ We corrected throughout the manuscript and figures “Claudin5” to “Claudin-5”.

Page 12, line 324: Change “Claudin5+/PLVAP- BBB competent state to a leaky Claudin5-/PLVAP+ state...” to “Claudin-5+/PLVAP- BBB competent state to a leaky Claudin-5-/PLVAP+ state...”.

→ We changed “Claudin5+/PLVAP- BBB competent state to a leaky Claudin5-/PLVAP+ state...” to “Claudin-5+/PLVAP- BBB competent state to a leaky Claudin-5-/PLVAP+ state...”.

Page 12, line 329: Change “...increased Claudin5 and LEF1 expression,...” to “...increased Claudin-5 and LEF1 expression,...”.

→ We corrected throughout the manuscript and figures “Claudin5” to “Claudin-5”.

Page 13, line 373: Check the following sentence „Interestingly, serum Netrin-1 levels increased in patients with neuroinflammatory Multiple sclerosis, EAE or type 2 diabetes^{52,54}.“ The syntax suggests that „EAE“ is also a pathological condition in human.

→ We agree with the reviewer’s comment and changed the following sentence “Interestingly, serum Netrin-1 levels increased in patients with neuroinflammatory Multiple sclerosis, EAE or type 2 diabetes” to “Interestingly, serum Netrin-1 levels increased in patients with neuroinflammatory multiple sclerosis or type 2 diabetes”

Page 16, line 456: Change „..., Caveolin1 (Cell Signaling, 3238S), VE-cadherin...” to „..., Caveolin-1 (Cell Signaling, 3238S), VE-cadherin...”.

→ We changed “...Caveolin1 (Cell Signaling, 3238S), VE-cadherin...” to “... Caveolin-1 (Cell Signaling, 3238S), VE-cadherin...”

Suppl Figure 2A: Change „Claudin5“ to „Claudin-5“.

→ We corrected throughout the manuscript and figures “Claudin5” to “Claudin-5”.

Suppl Figure 5B: Change „Claudin5“ to „Claudin-5“.

→ We corrected throughout the manuscript and figures “Claudin5” to “Claudin-5”.

Figure legend Suppl Figure 5: Change „Quantification of Claudin5 and PLVAP immunostaining...“ to „Quantification of Claudin-5 and PLVAP immunostaining...“

→ *We changed “...Quantification of Claudin5 and PLVAP immunostaining...” to “...Quantification of Claudin-5 and PLVAP immunostaining...”.*

Suppl Figure 8A: Change „Caveolin1“ to „Caveolin-1“, and change „ZO1“ to „ZO-1“.

→ *We changed “Caveolin1” to “Caveolin-1”, and “ZO1” to “ZO-1”.*

Reviewer #3 (Remarks to the Author):

The authors have addressed my concern with text changes, additional data and removal of unsubstantiated claims and limited data sets. The manuscript is markedly improved by the revision.

→ *We thank the reviewer for the critical assessment of our manuscript.*

Reviewer #4 (Remarks to the Author):

The authors addressed most of the comments raised by the reviewer. I do not have further comments or questions.

→ *We thank the reviewer for the critical assessment of our manuscript.*